# Intercontinental transport of biomass burning pollutants over the Mediterranean Basin during the summer 2014 ChArMEx-GLAM airborne campaign

Vanessa Brocchi[1], Gisèle Krysztofiak[1], Valéry Catoire[1], Jonathan Guth[2], Virginie Marécal[2], Régina Zbinden[2], Laaziz El Amraoui[2], François Dulac[3], Philippe Ricaud[2]

[1] LPC2E, CNRS – Université Orléans, F-45071 Orléans, France

[2] CNRM, Météo-France-CNRS, UMR 3589, F-31057 Toulouse, France

[3] LSCE/IPSL, CEA-CNRS-UVSQ, IPSL, Université Paris-Saclay, F-91191 Gif-sur-Yvette, France

*Correspondence to*: Gisèle Krysztofiak (gisele.krysztofiak@cnrs-orleans.fr)

**Abstract.** The Gradient in Longitude of Atmospheric constituents above the Mediterranean basin (GLAM) campaign was set up in August 2014, as part of the Chemistry-Aerosol Mediterranean Experiment (ChArMEx) project. This campaign aimed at studying the chemical variability of gaseous pollutants and aerosols in the troposphere along a West-East transect above the Mediterranean Basin (MB). In the present work, we focus on two biomass burning events detected at 5.4 and 9.7 km altitude above sea level (asl) above Sardinia (from 39°12N-9°15E to 35°35N-12°35E and at 39°30N-8°25E, respectively). Concentration variations in trace gas carbon monoxide (CO), ozone ($O_3$) and aerosols were measured thanks to the standard instruments on-board the Falcon-20 aircraft operated by the Service des Avions Français Instrumentés pour la Recherche en Environnement (SAFIRE) and the Spectromètre InfraRouge In situ Toute Altitude (SPIRIT) developed by LPC2E. 20-day backward trajectories with Lagrangian particle dispersion model FLEXPART (FLEXible PARTicle) help understanding the transport processes and the origin of the emissions that contributed to these pollutions detected above Sardinia. Biomass burning emissions came (i) on 10 August from the Northern American continent with air masses transported during 5 days before arriving over the MB, and (ii) on 6 August from Siberia with air masses travelling during 12 days and enriched in fire emission products above Canada 5 days before arriving over the MB. In combination with the Global Fire Assimilation System (GFAS) inventory and the Moderate Resolution Imaging Spectroradiometer (MODIS) satellite fire locations, FLEXPART reproduces well the contribution of those fires to CO and aerosols enhancements under adjustments of the injection height to 10 km in both cases, and application of an amplification factor of 2 on CO GFAS emissions for the 10 August event. The chemistry transport model (CTM) MOCAGE is used as a complementary tool for the case of 6 August to confirm the origin of the emissions by tracing the CO global atmospheric composition reaching the MB. For this event, both models agree on the origin of air masses with CO concentrations simulated with MOCAGE lower than the observed ones, likely caused by the coarse model horizontal resolution that yields the dilution of the emissions and diffusion during transport. In combination with wind fields, the analysis of the transport of the air mass documented on 6 August suggests the subsidence of CO pollution from Siberia towards North America and then a transport to the MB via fast jet winds located at around 5.5 km in altitude. Finally, using the ratio $\Delta O_3/\Delta CO$, the plume age can be estimated and the production of $O_3$ during the transport of the air mass is studied using the MOCAGE model.

**1 Introduction**

Biomass burning is a significant contributor of trace gas and aerosol content to the atmosphere (e.g. Andreae and Merlet, 2001). It emits large amounts of chemically active trace gases (e.g. carbon monoxide, CO and nitrogen oxides, $NO_x$) that impact the composition of the atmosphere at regional and global scales, with consequences on ozone ($O_3$) formation. CO, a tracer for biomass burning emissions produced from incomplete combustion, is a precursor of tropospheric $O_3$ and a sink for radical hydroxyl (OH), the main oxidising species in the atmosphere (Seinfeld and Pandis, 2016). Forest fires are also an important source of tropospheric aerosols that play a significant role on radiative properties of the atmosphere (Liousse et al., 1996) especially in the Mediterranean region (Pace et al., 2005; Bougiatioti et al., 2016).

Long-range transport is now recognized as one of the most important processes affecting the spatial variability of pollutants (Roiger et al., 2012). As a result, air pollution from one continent can alter the chemical composition of the atmosphere above another continent. Compounds from fires or photochemically produced within fire plumes can be transported horizontally, but also vertically and be injected into the mid to upper troposphere/lower stratosphere (e.g. Fromm and Servranckx, 2003; Colarco et al., 2004; Damoah et al., 2004; Jost et al., 2004; Nedelec et al., 2005; Damoah et al., 2006; Cammas et al., 2009; Dahlkötter et al., 2014) when the fire activity is strong enough and can thus affect climate and air quality (Val Martin et al., 2006 and references therein).

Looking at the projections of future climate change, the Mediterranean Basin (MB) is considered as highly sensitive and has been identified as one of the most important "hot-spots" (Giorgi and Lionello, 2008). It is at the crossroad of different transport processes (Lelieved et al., 2002; Millan et al., 2002; Gerasopoulos et al., 2005; Duncan et al., 2008; Doche et al., 2014; Ricaud et al., 2014) and at the intersection of different sources of pollutants, either natural (e.g. major dust sources from the Sahara and Arabian deserts) or anthropogenic, influencing the variability of aerosols (e.g. Nabat et al., 2013) and trace gases in the MB (Mihalopoulos, 2007). Within this scientific framework, Chemistry-Aerosol Mediterranean Experiment / Gradient in Longitude of Atmospheric constituents above the Mediterranean basin (ChArMEx-GLAM) campaign has been set up in order to document the variability in aerosol and trace gases in the free troposphere over the MB. Several studies have already reported transatlantic transport of boreal forests fire emissions from North America to Central Europe (Forster et al., 2001; Petzold et al., 2007) or to the MB (Formenti et al., 2002; Cristofanelli et al., 2013; Ancellet et al., 2016). Those studies were mainly focused on biomass burning aerosols except that of Forster et al. (2001) which also addressed CO emissions, but with measurements made in Germany, thus, concerning Central Europe. Damoah et al. (2004) and Spichtinger et al. (2004) have reported transport of fire emissions from Russia to Europe via eastward circumnavigation, but they were not specifically focused on the MB and did not include in-situ measurements at high altitude.

In the present paper, we analyse the intercontinental transport of CO and aerosol biomass burning emissions from North America and Siberia having impacted the MB based on two flights of the (ChArMEx-GLAM) airborne campaign on 6 and 10 August 2014. Only a limited amount of in-situ observations and trace gases distribution are available in the troposphere for the whole Mediterranean region (Di Biagio et al., 2015).

The in-situ measurements presented here, being direct, local and at high spatial resolution, are thus of interest for the region. First we describe in section 2 the GLAM aircraft campaign with the onboard instruments. The quality of the aircraft SPIRIT CO measurements is checked by comparison to a surface station located close to the Lampedusa airport where the aircraft landed and took off. Then, we explain the modeling work undertaken to characterize this long-range transport of biomass burning pollutants impacting the MB. Backward modeling with the Lagrangian particle dispersion model FLEXPART, and chemistry transport modeling with the 3D CTM MOCAGE are introduced. We address the question of emission inventories as well as the estimation of the injection height since they are sources of uncertainties when using models to determine long-range transport. In section 3, sensitivity tests of the models to the injection height and to the emissions are thus performed to properly simulate the fire products. To characterize the model sensitivity to the injection height, tracers are released at different altitudes, while for sensitivity to the emissions, the simulated concentrations are compared to the measurements. Once the emissions and the injection height are set up, model simulations with FLEXPART and MOCAGE are used to trace back the plumes source regions and to estimate the CO biomass burning contribution to the aircraft measurements, i.e. to the mid- (5.4 km above sea level (asl)) and upper (9.7 km asl) troposphere in the MB. A detailed analysis of the transport of the air mass from Siberia via North America to the MB on 6 August is also provided to conclude this section. Section 4 deals with the production of $O_3$ inside the plume during the air mass transport using the aircraft measurements and MOCAGE model. Section 5 presents the main conclusions.

## 2 Methodology

### 2.1 Campaign description

The GLAM aircraft campaign set up in the framework of ChArMEx aimed at studying the tropospheric chemical variability and trends of pollutants with different lifetimes above the MB during 6-10 August 2014. The campaign and the main results obtained are presented in Ricaud et al. (2017). To sum up, in-situ measurements of trace gases and aerosols were performed thanks to the instruments integrated in the Falcon-20 research aircraft operated by the French Instrumented Aircraft Service for Research in Environment (SAFIRE, CNRS and Météo-France). The 5-day long campaign consisted in 8 flights at various altitudes on the outbound and on the return leg between Toulouse (France) and Larnaca (Cyprus), via Menorca (Spain), Lampedusa (Italy) and Heraklion (Crete Island). During the first transect, from the western to the eastern part of the MB, the aircraft flew at an altitude level of 500 hPa (~5.4 km asl), also providing vertical profiles up to 12 km around each landing sites mentioned above. As calibrated measurement surface stations belonging to the World Meteorological Organization/Global Atmosphere Watch network (WMO/GAW) are located close to some landing sites, it was possible to compare aircraft and surface measurements. On the way back from the eastern to the western part of the MB, a constant pressure level of 300 hPa (~9.7 km asl) was held for the cruising altitude. This paper focuses on CO, $O_3$ and aerosol measurements during the flights on 6 August 2014 (flight F2) and on 10 August 2014 (flight F8).

## 2.2 Airborne payload

The SPIRIT airborne infrared absorption spectrometer for the measurements of trace gases was mounted onboard the Falcon-20. A complete description of the instrument principle can be found in Catoire et al. (2017). In brief, it uses continuous-wave distributed-feedback room-temperature Quantum Cascade Lasers (QCLs), allowing rapid scanning (each 1.6 s) of strong fundamental molecular ro-vibrational lines lying in the mid-infrared, with ultra-high spectral resolution ($10^{-3}$ cm$^{-1}$). The QCL infrared beams are absorbed by constituents of the ambient air sampled in a multipass cell (with 83.88 m of pathlength) at reduced pressure (33 hPa), and detected using a cooled HgCdTe photodetector. In the present campaign, measurements were carried out at the wavenumber 2179.772 cm$^{-1}$ for $^{12}C^{16}O$. Total molecule abundance is deduced from a home-made software using the HITRAN 2012 database (Rothman et al., 2013) with a precision of 0.3 ppbv for CO at 1.6 s time resolution. Dry volume mixing ratios (vmr) are deduced using the measured pressure and temperature of the optical cell, and using the water vapour mixing ratios measured by a laser absorption spectroscopy hygrometer (WVSS-II from SpectraSensors). Total uncertainties for CO were estimated to be 4.7 ppb using comparisons performed during previous flights with high altitude instrumented surface stations (Pic du Midi de Bigorre and Puy de Dôme, France) belonging to the World Meteorological Organization/Global Atmosphere Watch (WMO/GAW) network and with a NOAA standard cylinder on-board the aircraft (Catoire et al. 2017). They are also checked in the present campaign (see Section 2.3).

The Mozart instrument was used for the measurements of $O_3$ concentrations. It is a modified version of a commercial Ozone analyser (TEI 49C), an early version of the instrument used in the 'Measurement of OZone by Airbus In-service airCraft' (MOZAIC) program (Marenco et al., 1998). The measurement accuracy of Mozart is identical to the accuracy of MOZAIC instruments, which has been estimated at ±[2 ppbv + 2%] (Thouret et al., 1998).

The number of aerosol particles per cm$^3$ was provided by a Passive Cavity Aerosol Spectrometer Probe (PCASP-100X from Droplet Measurement Technologies, Inc) measuring the particles within a 0.1-3 µm size range into 30 bins. The principle of measurement is based on the scattering of light by individual particles going through a laser beam (visible wavelength). The particle size is determined from this particle light-scattering. The PCASP instrument is thus only able to determine size ranges but does not discriminate the type of aerosol. Thereafter, only the 0.2-1.1 µm size range is considered as this range size is representative of black carbon (BC) particles (Dahlkötter et al., 2014) and corresponds to the sum of the number of particles in 10 bins.

## 2.3 Comparison between the measurements and a WMO/GAW calibrated surface station

The presence of a calibrated measurement surface station (35.52°N, 12.63°E, 45 m asl) located at 2.5 km northwest from the Lampedusa airport represents an opportunity to confirm the quality of GLAM airborne measurements. Surface CO vmr are routinely measured at this station by a WMO/GAW calibrated Picarro instrument. This comparison is possible under the condition the air mass sampled is similar for both the station and the aircraft. Wind fields from ERA-INTERIM are 1) extracted at 6-hourly intervals (0000, 0600, 1200, 1800 UTC) and 3-hour forecasts (0300, 0900, 1500, 2100 UTC) with a horizontal resolution of 0.125°×0.125°, high

enough to specify the wind direction at the local scale, and 2) selected at the dates of the presence of the aircraft in the vicinity. Two landings and take-offs were performed on the outbound, on 6 and 7 August, and on the return leg, on 10 August. In the first case, the wind was blowing from northwest, thus the aircraft landed and took off downwind, while it was the contrary on 10 August (aircraft upwind). About 10 seconds of SPIRIT measurements are selected to respect the down/up wind condition within a band of 1 km width, corresponding to altitudes below

110 m asl, consistently within the maritime boundary layer (600 m asl). Results are gathered in Table 1 and show the excellent agreement between these CO measurements and those of the surface station: the difference (-3.5 to +5.1 ppb) observed is within the total estimated uncertainties reported for both instruments (4.7 to 7.9 ppb). For the take-off on 7 August, the SPIRIT was not ready to make such comparison.

## 2.4 Lagrangian particle dispersion modeling

The Lagrangian transport and diffusion model FLEXPART (FLEXible PARTicle; Stohl et al., 2005) is used in our study to describe the transport of air masses to the MB. It simulates long-range transport, diffusion, dry and wet deposition of atmospheric tracers by computing trajectories of a large number of particles. It calculates the trajectories of released particles, taking into account advection and turbulent diffusion processes.

Model calculations are based on meteorological data from the European Centre for Medium-Range Weather

Forecasts (ECMWF). ERA-INTERIM (Dee et al., 2011) meteorological re-analysis data provided by ECMWF have been used for all simulations. Data are extracted at 6-hourly intervals (0000, 0600, 1200, 1800 UTC) and 3-hour forecasts (0300, 0900, 1500, 2100 UTC) with a resolution of 0.5°×0.5° in latitude and longitude, a compromise at the global scale between computing cost and the trajectory accuracy. A vertical resolution of 137 model hybrid levels is used with the model top at 0.01 hPa.

Our FLEXPART calculations are performed backward in time (Seibert et al., 2004) with the model version 9.0. Possible source contributions to the content of the air mass are determined by releasing particles from points located at the flight track. This backward mode gives access to two products. The first one is a residence time of the particles in the total atmospheric column while the second one, defined as the potential emission sensitivity (PES), informs on the location where the sampled air mass has been impacted by the surface

emissions. The emission sensitivity refers to the source-receptor relationship (Seibert et al., 2004). It describes the sensitivity of a receptor (here, the aircraft) to a source of emissions. The thickness of the PES layer is chosen consistently with the altitude (vertically integrated values from 0 to 10 km; see below) at which the emissions are injected in order to take into account the injection of fires in the mid to upper troposphere. Outputs are averaged every 24 hours with a horizontal resolution of 0.5°×0.5° globally. Retroplume trajectory outputs are condensed

into a cluster analysis (Stohl et al., 2002b).

      In the present case, the model simulations are run with the CO tracer, only considering reaction with OH radicals (with concentrations monthly-averaged with 3°×5° resolution from GEOS-CHEM model; Technical note FLEXPART v8.2, http://flexpart.eu/downloads/26) and with the aerosol tracer assuming removal by dry and wet depositions with properties similar to BC. Each simulation consists of 20,000 particles released at the aircraft

location and time of measurements. They are followed during 20 days backward in time with parameterization of

turbulence, activation of the convection, and age of air calculation ("lagespectra" option). The convection is based on the Emanuel and Zivkovic-Rothman (1999) scheme that only relies on the ECMWF temperature and humidity fields.

In addition, the Global Fire Assimilation System (GFAS) version 1.2 emission inventory (Kaiser et al., 2012, http://apps.ecmwf.int/datasets/data/cams-gfas/) is used for CO and aerosols for year 2014 with a 0.1°×0.1° grid. GFAS calculates biomass burning emissions by assimilating fire radiative power (FRP) observations from MODIS instruments (MOderate resolution Imaging Spectroradiometer; Giglio et al., 2003; http://modis-fire.umd.edu/pages/ActiveFire.php) onboard the Terra and Aqua satellites. This FRP gives quantitative information on the combustion and smoke emission rates (Ichoku and Kaufman, 2005; Wooster et al, 2005). Multiplying the CO emission flux from this inventory with the FLEXPART emission sensitivity gives access to the contribution of biomass burning sources to the total CO abundance (in ppb m$^{-2}$) present in the air sampled by the aircraft. For aerosol simulations, a multiplicative coefficient of 3.4 is applied to the GFAS emissions as recommended in Kaiser et al. (2012). As for CO, aerosol emission flux multiplied with FLEXPART emission sensitivity gives access to the contribution of biomass burning sources to the total aerosol abundance (in ng m$^{-3}$).

**2.5 Chemistry-transport modeling**

MOCAGE (MOdèle de Chimie Atmosphérique à Grande Echelle; Josse et al., 2004; Sič et al., 2015; Guth et al., 2016), version R2.15.0, is used in this study to simulate the atmospheric composition during July and August 2014. In particular we analyse the CO global atmospheric content in order to trace the biomass burning events and the concentration and production of $O_3$ inside the plume. MOCAGE is a 3D stratospheric and tropospheric Chemistry Transport Model (CTM) using a semi-Lagrangian transport scheme.

In order to represent both the tropospheric and the stratospheric air composition, two chemical schemes are implemented into MOCAGE. The Regional Atmospheric Chemistry Mechanism (RACM) (Stockwell et al., 1997) is used in the troposphere. For the stratosphere, it is the REPROBUS scheme (REactive Processes Ruling the Ozone BUdget in the Stratosphere) which is implemented (Lefèvre et al., 1994).

MOCAGE is an off-line model and thus needs external meteorological forcing based on wind and temperature fields from the analyses of the operational model of Météo-France, ARPEGE (Courtier et al., 1991). MOCAGE can be run with several nested grids. In our configuration, the horizontal resolution for the global domain is 2°×2° with 47 vertical levels. A regional nested domain, defined with a horizontal resolution of 0.2°×0.2°, is centered over the MB. For the global domain, the Global Emissions InitiAtive (GEIA; Guenther et al., 1995) and the MACCity inventories are used (Lamarque et al., 2010; Granier et al., 2011; Diehl et al., 2012) for natural and anthropogenic emissions, respectively. For the regional domain, anthropogenic emissions are taken from the MACC II (Monitoring Atmospheric Composition and Climate - Interim Implementation) inventory (Kuenen et al., 2011), biomass burning emissions are from the Global Fire Assimilation System GFAS 1.1 product (Kaiser et al., 2012) while natural emissions are from the GEIA inventory. Those emissions can be found at eccad.sedoo.fr.

## 2.6 Remote sensing products

The Cloud-Aerosol Lidar with Orthogonal Polarization (CALIOP Lidar) is a two-wavelength polarization-sensitive Lidar that provides high resolution vertical profiles of clouds and aerosols (Winker et al., 2009). With also an Imaging Infrared Radiometer (IIR) and a Wide Field Camera (WFC), CALIOP is onboard the Cloud-Aerosol Lidar and Infrared Pathfinder Satellite Observations (CALIPSO), a component of the A-Train constellation, launched on 28 April 2006. In our study, we use the 532-nm total (parallel and perpendicular) attenuated backscatter aerosol (https://www-calipso.larc.nasa.gov/products/lidar/) to determine at which maximum altitude smoke is detected.

The Atmospheric Infrared Sounder (AIRS) instrument was launched aboard the second Earth Observing System (EOS) polar-orbiting platform, EOS Aqua, in May 2002. It is a hyperspectral infrared grating spectrometer. Its goal is to support climate research and improve weather forecasting by observing and characterizing the entire atmospheric column from the surface to the top of the atmosphere. We use CO zonal means (mean area is represented in Figure 4; https://giovanni.gsfc.nasa.gov/giovanni/) from AIRS instrument with a daily and $1°\times1°$ resolution. These data are used to identify the CO injection from fires.

## 3 Analysis of the long-range transport of biomass burning encountered over the MB during the GLAM flights

This section presents the modeling work associated to the aircraft observations. With FLEXPART, the method used to determine the geographic origin of the pollution and the sources of emission is the same in both case studies, i.e. we use 20-day back-trajectories in order to have access to the PES (see Section 2.4). The PES map informs on the location where the air mass has taken up emissions. We then combine the PES with CO and BC emissions from GFAS inventory and derive the biomass burning contribution to CO and aerosols measured concentrations.

### 3.1 Case study of 10 August 2014: Northern American forest fire plume

The F8 flight (8[th] flight of the campaign, on 10 August) consists of an east-to-west flight from Lampedusa (Italy) to Toulouse (France) (Fig. 1a and 1b). During the transect at about 9.7 km asl, an increase of CO vmr up to ~110 ppbv (from a background at ~70 ppbv) has been measured above Sardinia. A very intense and transitory increase of CO up to about 260 ppbv has been measured among this general increase of CO, correlated with a weaker increase in $O_3$ (from ~35 ppbv to ~75 ppbv) and aerosols up to about 1000 particles cm$^{-3}$ in the 0.21-1.1 µm diameter range, and a decrease in relative humidity (RH). A picture taken during the flight shows that the aircraft traversed a thin dark layer of particles (Fig. 1c). This intense peak lasted about 10 minutes as the aircraft was flying from Lampedusa back to Toulouse, and so passed rapidly through the pollution layer. This event has already been evoked in Ricaud et al. (2017). They used 20-day back-trajectory calculations of Hybrid Single Particle Lagrangian Integrated Trajectory (HYSPLIT) model with global reanalysis data on a $2.5°\times2.5°$ grid to assess the origin of the air parcels. It was shown that air masses originate from the Northern Territories (Canada)

above 7 km and the United States below 7 km, where fires were detected by MODIS. MOCAGE simulations confirmed this result and found elevated amounts of CO and BC over North America. In our study, we use FLEXPART model with a thinner resolution (0.5°×0.5°) during 20 days backward in time to reproduce this intense peak. FLEXPART allows to assess the origin of the air masses but also to calculate the biomass burning contribution of CO and BC to the measurements by coupling it with GFAS inventory. A direct comparison between the simulated and measured concentrations is thus possible. Thanks to the calculation of the age of air, it is also possible to discriminate the different contributions in terms of dates before the flight.

To determine the origin of this pollution, 20-day backward trajectories are calculated using FLEXPART all along the flight track. Emission sensitivities are stored on a 3-D grid with levels from the surface up to 10 km asl. The representation of the PES is a good indicator of where and for how long the air mass has probably taken up emissions.

20-day back-trajectories (Fig. 2a) originated above Sardinia where CO is at its maximum (13.23h UTC) reveal that the air masses stayed mostly above Mongolia and Northern China between 12 to 19 days before the flight (from 22 to 26 July). Then, the air masses traveled eastward where they passed above central Pacific between 6 to 10 days (31 July to 4 August) before the flight and above Canada and the United States 4 to 5 days (5 to 6 August) before reaching Sardinia. As shown in Fig. 2a, numerous fires were detected by MODIS for more than one month in Northwestern Canada and the United States and especially intense fires triggered by lightning ignition and drought in California and Oregon (see https://earthobservatory.nasa.gov/IOTD). The confidence for every MODIS fire detection is characterized by a number ranging 0-100%. We only use fire detections with a confidence level greater than 75%. The map of the CO contribution from biomass burning (Fig. 2b) from FLEXPART reveals that the strongest contribution comes from Siberia, Northwestern Canada and the West Coast of the United States (the last two gathered for simplicity under one single name, the Northern American continent), but almost no biomass burning contribution comes from Mongolia and China. To exclude the possibility of anthropogenic contribution by China, we coupled FLEXPART with EDGAR (Emissions Database for Global Atmospheric Research; EC-JRC/PBL, 2011) v4.2 inventory. EDGAR provides country- and sector-specific anthropogenic emissions of greenhouse gases and pollutants (including CO). The CO anthropogenic contribution map (not shown) reveals that Chinese contribution is less than 10 ppbv and thus has a negligible influence on the CO values measured. As illustrated in Figure 3a, CO contribution simulated by FLEXPART reproduces well the measured CO, with a slight delay of only 3 min. We isolated the CO emissions of the main areas of biomass burning contributions as shown in Figure 2b. This leads to the conclusions that the Northern American continent is the only source of CO emission as the contribution of Siberia is close to zero (Fig. 3a). Considering the age of the air mass (Fig. 3b), we note that the maximum of contribution is 4 to 5 days (namely 5 and 6 August) before the flight when the PES was above the Northern American continent in consistency with the above result.

To quantitatively compare the two CO datasets (simulated by FLEXPART and measured by SPIRIT), a background value of 70 ppbv was added to the FLEXPART data. This added value of 70 ppbv is an average of our measurements during the campaign, thus representative of the CO background of the mid-troposphere for the western MB at that period. Although FLEXPART is able to simulate the origin of this pollution, we note (Fig. 3a)

that the CO calculated vmr are less than the CO measurements. Since the effect of pyroconvection is not included in FLEXPART, different scenarii are tested in order to reproduce the CO impact of those fires. Thus, we adjusted the injection height of the plume as it plays an important role in its long-range transport. As a matter of fact, the

consequences on the transport, the deposition and the lifetime depend on whether the plume is below or above the planetary boundary layer. The injection height depends on several variables as the intensity of the fire, but also on the synoptic conditions. If the meteorological conditions are satisfied, i.e. water vapour condensation and release of latent heat due to an environment cooler than the plume, then pyroconvection can occur (Fromm et al., 2010). To determine the initial injection height, we use data from CALIOP Lidar and AIRS instrument. CALIOP and

AIRS data are selected for the same day and with close geographical coordinates enabling the comparison between the two datasets. The nighttime CALIOP overpass above North America on 6 August at 9:51 UTC can be used to locate the fire plumes (Fig. 4a left). The aerosol subtypes indicate the presence of smoke up to 10 km asl from latitudes 38-62°N (Fig. 4a middle). For the same day and from latitudes 61-64°N, high CO concentrations from AIRS (Fig. 4a right) can be found up to 10 km with maximum concentrations above 5 km.

An estimation of injection height derived from MODIS instrument data and meteorological information from ECMWF is also provided by GFAS. GFAS estimations of injection height are on 5 and 6 August at a maximum of 10.9 km above Canada (not shown).

Other studies have already proved that the use of such an altitude of 10 km to inject surface emissions in the atmosphere allows simulating concentrations in good agreement with the measurements (De Gouw et al.,

2006; Elguindi et al., 2010). However, the measured CO concentrations are still about twice higher than the simulated ones (Fig. 3a). As FLEXPART has already proved its accuracy in simulating events of long-range transport of fire plumes (Forster et al., 2001; Damoah et al., 2004, 2006; De Gouw et al., 2006; Stohl et al., 2007; Lapina et al., 2008; Cristofanelli, 2013) and the question of convection induced by fires has been solved by applying a higher injection altitude, one hypothesis for those lower concentrations would be an underestimation

of the GFAS CO emissions, although Kaiser et al. (2012) do not discuss a correction to be applied to CO emissions. As errors depend on individual fire events because the fire space-borne observations depend on the instrument sampling, as e.g. cloud-free observations (J. Kaiser, pers. com.), the underestimation detected in our study cannot be considered as a general statement that has to be applied in every case study using GFAS CO emissions, but only in ours. Finally, an amplification factor of 2 has to be applied to get similar CO quantities

during the event, derived from the calculated surface under the CO enhancement (Fig. 3a).

The FLEXPART simulations for the biomass burning contribution to the total BC use the same parameters as for CO simulations, in particular an injection height up to10 km. In Figure 3c, FLEXPART is able to qualitatively reproduce the fine aerosol concentration peak measured by the PCASP, delayed by ~3 minutes as for CO. The contribution map gives the same regions, i.e. Northern American continent and Siberia, to the

aerosol loading (not shown). To distinguish which region contributes the most to the peak of pollution, we isolate the BC contribution to the measurements in each of these areas. It appears that the Northern American continent is the main source of BC emissions and that Siberia contributes for less than 3 ng m$^{-3}$ (Fig. 3c).

The injection of such a quantity of CO and aerosol particles measured at high altitude (>9 km) is therefore due to a pyroconvective lifting that has uplifted pollutants at high altitude, which subsequently traveled

over the North Atlantic. Fire products in North America transported to Europe and more specifically to the MB seem to be recurrent but of variable intensity. Pu et al. (2007) found, thanks to satellite data, that most of the fires occur in June-July in North America. Stohl et al. (2002a), also using FLEXPART simulations, found that the MB is the location where the highest surface concentrations of the Northern America tracers are detected in summer. Different corridors of pollution impacting the MB, including the one from North America, have been identified in

Ricaud et al. (2017), without discriminating the biomass burning contribution to CO concentrations. Transatlantic transport is well documented and comparable transport durations between North America and Europe are found in other studies: about 6 days and 7 days in Petzold et al. (2007) and in Forster et al. (2001), respectively. Transport between North America and the MB in Formenti et al. (2002) and Ancellet et al. (2016) lasts about 10 days (Table 2). A hypothesis of Forster et al. (2001) is that those biomass burning emissions coming from North

America might have an influence on European pollutant concentration levels during the summer period almost every year, but with variable intensity each year. Over Central Europe, Forster et al. (2001) and Petzold et al. (2007) have found a fire plume layer up to 6 km and 8 km asl, respectively. Over the MB, Formenti et al. (2002) have found a forest fire haze layer up to 3.5 km coming from Canadian forest fires over the eastern Mediterranean. Ancellet et al. (2016) have reported aerosol layers up to 7 km coming from North America over

the western MB. The published cases we know of transatlantic transport to MB in the recent years are gathered in Table 2. Fromm et al. (2005) and Damoah et al. (2006) also report pollution from pyroconvective fires over North America. Moreover, they indicate that the fire activity was strong enough to inject smoke up into the lower stratosphere.

### 3.2 Case study of 06 August 2014: Siberian biomass burning

The flight F2 (2nd flight of the campaign, on 06 August) consists of a west-to-east flight from Menorca (Spain) to Lampedusa (Italy), as shown in Figure 5a. It is characterized by a transect at about 5.4 km asl with a vertical profile above Lampedusa up to 12 km. During this transect (Fig. 5b), we measured above Sardinia an enhancement of CO from ~70 ppbv to ~120-140 ppbv at 12:00 UTC, synchronized with enhancements of $O_3$ from ~30 to ~60 ppbv, of aerosols up to about 100 particles cm$^{-3}$ in the 0.21-1.1 μm size range and a decrease in

RH, that lasted more than 40 minutes (~ 509 km traveled). The background concentrations are rather similar to the ones measured during F8, however the peak intensity of CO is lower. The aircraft was flying in the same direction as the air mass motion (see section 3.2.2 below) which could explain why it stayed so long in the polluted air mass. After that, the aircraft performed an ascending vertical profile, and crossed again this layer at the same altitude and with the same features more than 30 minutes later (just before 13:30 UTC) when

descending for landing on Lampedusa island. The measurements performed during this vertical profile help us determining that the thickness of the layer is 2.9 km.

### 3.2.1 Analysis of the origin of the air mass using FLEXPART

The PES map shown in Figure 2c reveals that the majority of the air masses were above western Canada about 5 days before (1 August) and close to Siberia, Northwestern China and western Mongolia about 12 to 16 days (21 to 25 July) before the flight. Along these trajectories, MODIS instrument on Terra and Aqua satellites detected fires on those dates mostly in Northwestern Canada and Siberia (Fig. 2c). The contribution map presented in Figure 2d indicates that source contributions to CO are located mainly in Northwestern Canada and in Siberia.

As in the previous case (see section 3.1), we have to inject forest fire emissions at high altitudes in FLEXPART simulations in order to mimic the vertical transport of fire plumes. CALIOP overpasses above Siberia and North America are used to estimate this injection height. The first CALIOP profile, shown in Figure 4b, taken above Siberia on 24 July at 18:37 UTC indicates smoke detected up to 10 km altitude for latitudes from 47°N to 59°N, with most of the aerosols around 5 km. The second profile (Fig. 4c) above the eastern coast of North America on 2 August at 6:58 UTC shows smoke detected up to 11 km at latitudes from 59 to 65°N with most of the aerosols detected above 5 km. AIRS data are also used to identify the CO injection from fires. On 24 July (Fig. 4b), high CO concentrations are detected up to 10 km, with maximum concentrations above 5 km from latitudes 50-60°N. On 2 August (Fig. 4c), CO concentrations are also detected up to 10 km with maximum values above 5 km from latitudes 60-70°N. CALIOP and AIRS are both in agreement to estimate the injection height of the plume. GFAS estimations of injection height on 24 and 25 July are 7.9 km above Siberia, 10.4 km on 1 August and 9 km on 2 August above Canada (not shown). Two injection heights are tested in the model above Siberia and Canada for this case: at 6 and 10 km. For Siberia, the difference between the two simulations at 12.0-12.8h UTC is 7%, meaning that most of the emissions emitted by fires are at altitudes between 0 and 6 km even though some can reach higher altitudes. For Canada, the difference for the same period of time is much higher (27%), with the altitude of 10 km giving the strongest contributions. By using an injection height of 10 km for both regions, the GFAS global contribution to CO vmr calculated by FLEXPART simulates well the measured peak of pollution between 12.0h and 12.8h UTC, as illustrated in Fig. 6a. In order to verify that the global CO anthropogenic contribution does not affect our results, a coupling between FLEXPART and EDGAR v4.2 inventory is performed. The contribution map (not shown) shows that CO anthropogenic contribution accounts for less than 10 ppbv. Selecting each area to find where the biomass burning contribution dominates shows that, in the case of CO, Siberia and Canada contribute approximately in the same manner but with a slightly higher contribution from Siberia (Fig. 6a). The biomass burning contribution in Southern Russia (North of the Black Sea) has also been studied but leads to a contribution close to zero (not shown). Looking at the age of the air mass (Fig. 6b), two main contributions are emphasized between 12.0 h and 12.8 h UTC: the first maximum is between 1 and 2 August (i.e. 5 and 4 days before the flight) when the PES was above Canada and the second maximum is between 24 and 25 July (i.e. 13 and 12 days before the flight) when the PES was above Siberia.

For the simulations of BC, the contribution map for an injection height of 10 km shows that Canada and Siberia are the main sources of emissions as well (not shown). The loading of aerosols from 12.0 h to 12.8 h UTC simulated by FLEXPART is correlated to the measurements, as illustrated in Figure 6c. Two peaks of aerosols, not concomitant with CO, are also measured around 13.1 h and 13.4 h UTC (Fig. 5). FLEXPART simulation

shows that these peaks are not related to biomass burning as it is not reproduced by the model (Fig. 6c). These

two spikes of aerosols could be considered as being dust particles as they also appear in the 1.1-3.1 µm size range measured by the PCASP instrument (not shown). Concerning the BC contribution from 12.0 h to 12.8 h UTC for each area, Canada is the main contributor for the whole peak (Fig. 6c). BC emitted by the Siberian fires 13 to 12 days before the flight traveled longer than the BC emitted by Canadian fires. Wet and dry removal is activated in FLEXPART simulations. Both phenomena closely depend on the particle size and on humidity and temperature

from the meteorological input data for the wet deposition. These phenomena induce short lifetime for the particles from a few hours to a few days (Bond et al., 2013), and explain why Siberia does not contribute significantly to the aerosol content sampled by the aircraft for F2.

Trans-Pacific transport of CO and particles from the Asian continent to the northeast Pacific is well documented (e.g. Bertschi et al., 2004 and references therein; Jaffe et al., 2004; Spichtinger et al., 2004; Heald et

al., 2003, 2006 and references therein; Bertschi and Jaffe, 2005; Holzer et al., 2005) with a transport within 5 to 8 days and with plumes at altitudes ranging from the mid to the upper troposphere. Boreal forest fires, most frequent in Canada and Siberia (Spichtinger et al., 2004), occur mainly from May to October (Lavoué et al., 2000). The location of Siberian emissions is in agreement with Liang et al. (2004) who indicate that summertime emissions from boreal forest fires are exported at latitudes higher than 55°N, what is confirmed by Spichtinger et

al. (2004) with the highest concentrations above Siberia found at latitudes up to 70°N. Extreme CO concentrations (as high as 800 ppbv) were measured around Lake Baikal at 8.0-12.5 km altitude (Nedelec et al., 2005). Such an event with eastward circumnavigation starting from Russia to Europe is reported in Damoah et al. (2004). In this study, the plume simulated by FLEXPART over Alaska was found at altitudes between 2 and 5 km and a little higher above Canada, between 4 and 7 km. In Spichtinger et al. (2004), eastward transport of

Siberian fire emissions in 1998 towards Canada followed by transport towards Europe is reported. There again, the plume over Canada was found at higher altitude (2-8 km) than the one above the Siberian source region (0-6 km).

To conclude, about 12 to 13 days before our flight F2, the air masses originated from Siberia where they have been loaded in CO and aerosols because of boreal forest fires. The smoke plume traveled eastward to

Canada where the air masses have been loaded again in pollutants because of fires in this region. They then crossed the Atlantic at 50°N and went southeastward off European coasts up to the western MB where they were detected by our aircraft instruments on 6 August at ~5.4 km altitude above Sardinia.

**3.2.2 Analysis of the transport of the air mass to the MB using MOCAGE**

To complete the analysis performed with back-trajectories from FLEXPART simulations, 3D MOCAGE Chemistry Transport Model (CTM) simulations of the global atmospheric composition for the biomass burning event reaching the MB on 6 August (F2) are performed. This allows us to confirm the origin of the pollution using a complementary tool. As in the FLEXPART simulations, the biomass burning emissions, taken from the GFAS inventory, are set up to an injection height of 10 km. Meteorological fields are taken from the operational

analysis of the ARPEGE numerical weather prediction model used at Météo-France. The dynamics is then

slightly different from the one used in FLEXPART simulations. Figure 7 presents the CO concentration (in ppbv) at 5.5 km altitude between 23 July (a) and 6 August (o) at 12h UTC. The red ellipses correspond to the position of the air mass from 23 July to 6 August reaching the MB at the time of the observations during F2 above Sardinia. This Figure is to be compared with Figure 2c, representing the footprint for the 20-day backward simulation made with FLEXPART relative to this episode.

There are biomass burning emissions between 23 and 26 July (Fig. 7a to 7d) over Siberia. A part of these emissions is transported towards the Northern American continent. This transport starts with a branch going North over the Bering Strait before doing a curl over the Pacific Ocean off the coast of Canada between 26 and 30 July (Fig. 7d to 7h). This behavior is consistent with the FLEXPART footprint in Figure 2c showing a curl over the Pacific Ocean.

Between 31 July and 3 August (Fig. 7i to 7l), there are biomass burning emissions over North America which get mixed with the biomass burning plume coming from Siberia. Figures 8a and 8b present cross sections of the mean CO concentration and the vertical wind in a box between 40°N and 70°N for 1 August 12UTC (the area is represented in green in Figure 7a), respectively. Figure 8a shows the track of the CO coming from Siberia (between 130°W and 100°W) mixing with the fresh emissions over North America (around 95°W). This Figure suggests a descent of the air mass including CO transported from Siberia towards North America where biomass burning fires occur. This idea is supported by the vertical wind speed represented in Figure 8b where positive values are a sign of subsidence. During this period (Fig. 7i-k), one can see a separation over the Northern American continent with a branch passing over northern Canada and a second one over the United States. This is also consistent with the FLEXPART footprint which shows a preferential trajectory for the northern branch.

Between 3 and 6 August (Fig. 7l to 7o), the pollution measured during F2 traveled around 5.5 km height from the East coast of the Northern American continent to the MB via the fast jet winds (Fig. 8c) which represent the mean wind field at 5.5 km between 3 and 6 August. These strong winds at 5.5 km correspond to the lower part of a high altitude jet located around 10 km height (not shown). This travel of the plume over the Atlantic Ocean is similarly found by the FLEXPART backward simulation (Fig. 2c).

At the time and location of the measurements, the simulation made with the model MOCAGE gives CO vmr around 90 ppbv in the plume and 60 ppbv in the area around (Fig. 7o). Even if these values are lower than the measured ones, MOCAGE reproduces the variation of CO inside and outside the plume. The horizontal resolution of the model, being 2°×2°, induces a dilution of the emissions and then a diffusion during the transport of the plume over several days. The FLEXPART simulation shows a slightly dominant contribution for the Siberian part of the fire emissions compared to North America (Fig. 6a). The numerical estimation of the contribution from the Siberian and the Northern American fire emissions was not directly calculated by MOCAGE. However, according to the modeled concentrations, the contribution of the Siberian part seems to be less than the contribution of the Northern American part. The different dynamical fields used for the simulations (ARPEGE for MOCAGE and ERA-INTERIM for FLEXPART) can produce differences in the results. These can also be explained by the model configuration, since FLEXPART is a Lagrangian model, while MOCAGE is a 3D CTM using a semi-Lagrangian transport. The geometry of MOCAGE with coarse resolution along with the

model diffusion makes the longer transport of the Siberian emissions more diluted than the Northern American emissions.

**4 Analysis of O$_3$ production during long-range transport**

**4.1 Plume age calculation by the aircraft measurements**

The simultaneous increase of CO and O$_3$ measurements shows the production of O$_3$ inside the plume (Fig. 1 and 5). The ratio $\Delta O_3/\Delta CO$ for the increase of the species with respect to their background values averaged over 20 minutes before and after these increases for F2 and F8, respectively, is of about 0.25 for flight 8 and of about 0.50 for flight 2. It has been shown that this ratio increases with the age of the plume (Jaffe and Widger, 2012). For our two flights and for boreal regions, these ratios correspond to a plume age $\geq$ 5 days (Jaffe and Widger, 2012; Parrington et al., 2013; Arnold et al., 2015). More precisely, the ratio gives an approximate plume age of 6-10 days for F8 and of 13-15 days for F2 (Jaffe and Widger, 2012), in agreement with the age of the air mass calculated with FLEXPART.

**4.2 Analysis of O$_3$ production with MOCAGE**

In this section, MOCAGE simulation is used to analyse the O$_3$ production inside the biomass burning plume during long-range transport. For flight F2, the emissions are set up to an injection height of 10 km without any coefficient applied to the emissions. MOCAGE simulates fairly well the O$_3$ background that is of ~40 ppbv compared to ~32 ppbv for the measurements (not shown). The simulation reproduces the variability of O$_3$ in good agreement with the measurements. For the first period of interest, between 12.0 h and 12.8 h UTC, MOCAGE simulates an increase of ~25 ppbv O$_3$ compared to ~35 ppbv for the measurements. For the second period of interest, at about 13.5 h UTC, MOCAGE simulates an increase of ~30 ppbv compared to ~50 ppbv for the measurements. Note that MOCAGE provides smoother peaks than the observations because of the finer resolution of the observations compared to the model. Considering this, MOCAGE reproduces well the measurements of flight F2 and is thus used to study the production of O$_3$ along the transport.

Figure 9 shows both the O$_3$ vmr and O$_3$ production on 25 July and 1 August at 5.5km in altitude. The complete panels of maps from 23 July to 6 August are provided as supplementary material to follow the production (Fig. S1) and the concentrations (Fig. S2) of O$_3$ during the travel of the air mass from Siberia to the MB. It shows high O$_3$ production in the biomass burning plume up to 3 days after the emission (Fig. S1). After that, the ozone production is lowered indicating an aging of the air mass. On 25 July, the production of O$_3$ is visible above Siberia between 40°N and 70°N (Fig. 9a). Figure 9b shows this production of O$_3$ with concentrations of O$_3$ greater than 110 ppbv in the same area. Then, the air mass crosses the Pacific Ocean before arriving above Canada. On 1 August, the simulation shows the production of O$_3$ between 30°N and 90°N (Fig. 9c). The concentrations of O$_3$ in Figure 9d are more important in this area, especially around 45°N with concentrations up to ~100 ppbv and around 70°N with concentrations up to more than 120 ppbv.

**5 Conclusions**

This study describes two remarkable events of long-range transport of biomass burning above Sardinia in the western Mediterranean during the ChArMEx-GLAM airborne campaign in August 2014. The in-situ measurements of the campaign contribute to expand the few available in-situ observations needed to better describe the trace gases distribution over the MB. In one case (F8, 10 August) a very intense peak of CO of about 260 ppbv was measured at about 9.7 km asl, correlated to an $O_3$ and aerosol peak and a decrease in RH. In the

other case (F2, 6 August) an enhancement of CO, $O_3$ and aerosols as well as a decrease in RH were measured at about 5.4 km asl and lasted for more than 40 minutes along the flight. We make use of in-situ measurements of CO and aerosols that we combined with models to analyse the intercontinental transport. The origin of those events is studied thanks to the Lagrangian transport model FLEXPART. Using the potential emission sensitivity maps and selecting specific areas applied to the contribution maps show that, in the 10 August episode,

emissions from the Northern American continent traveled during about 5 days before being measured in the MB and that, in the 6 August episode, biomass burning emissions from Siberia circumnavigated eastward over the globe before arriving above the Canada where the air mass was loaded again in biomass burning emissions and finally impacted the MB free troposphere. In this last case, the 3D CTM MOCAGE confirms this transport from Siberia towards North America and then towards the MB. FLEXPART was able to reproduce BC and CO

biomass burning contribution after having adjusted the two following parameters: the injection height and the amplification of the CO emissions from GFAS inventory. The change in injection height can be explained by the fact that pyroconvection is not taken into account in the model. The choice of this altitude of injection is validated with CALIOP and AIRS data and GFAS estimations of injection height. Finally, a height of 10 km is set in both cases. MOCAGE, with an injection height set at 10 km as well, qualitatively reproduces CO and $O_3$ concentration

variations. A detailed study of the transport of the air mass for the 6 August case reveals a subsidence leading to a descent of CO from Siberia towards North America. Then, this pollution travels from the East coast of the Northern American continent to the MB at an altitude of about 5.5 km, in the lower part of a high altitude jet. Throughout of this transport, MOCAGE simulates fairly well the production of $O_3$ inside the plume and in particular in the vicinity of the fire emission sources.

It would be interesting in the future to perform specific simulations in order to quantify the frequency of occurrence of those long-range transport events of high latitude summer forest fires.

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

**Author contributions**

F. Dulac coordinated the ChArMEx program and P. Ricaud was the initiator and coordinator of the GLAM campaign. P. Ricaud, R. Zbinden and V. Catoire participated in the SPIRIT measurements onboard the Falcon-20 and performed flight data analyses. FLEXPART simulations were performed by G. Krysztofiak and V. Brocchi. The study of air mass transport and regional chemistry modeling was conducted by J. Guth, V. Marécal and L. El Amraoui. V. Brocchi wrote the manuscript with contribution from all co-authors.

**Acknowledgements**

The authors thank S. Chevrier and C. Robert for conducting SPIRIT measurements and for their instrumental support during the campaign. The SAFIRE's crew (Service des Avions Francais Instruments pour la Recherche en Environnement) is acknowledged for flying operations and B. Piguet for the aircraft instrumental data processing. Measurements at Lampedusa by ENEA were partly supported by the Italian Ministry for University and Research through the NextData and Ritmare projects. We thank Sferlazzo di Sarra and Piacentino di Ioro for providing the dataset used in this study. The AIRS project is supported by the NASA Earth Observing System Program. CO from AIRS data were obtained from the NASA Giovanni system. We would like to thank the Centre National de la Recherche Scientifique – Institut National des Sciences de l'Univers (CNRS-INSU), Centre National des Etudes Spatiales (CNES), Agence de l'Environnement et de la Maîtrise de l'Energie (ADEME), Météo-France, and the Commissariat à l'Energie Atomique et aux Energies Alternatives (CEA) that funded the Chemistry-Aerosol Mediterranean Experiment (ChArMEx) as part of the programme Mediterranean Integrated STudies at Regional And Local Scales (MISTRALS). This work was funded by the Labex VOLTAIRE (ANR-10-LABX-100-01) and the PIVOTS project provided by the Région Centre – Val de Loire (ARD 2020 program and CPER 2015 -2020).

| Date | Aircraft location | CO vmr from the surface station (ppb) | CO vmr from the aircraft (ppb) | Difference in CO vmr between the surface station and the aircraft |
|---|---|---|---|---|
| 6 Aug, 13:52 UTC | 35.501°N – 12.638°E 84 ± 21 m asl | 115.9 ± 5.6* | 119.4 ± 4.7 | -3.5 ± 10.3 |
| 10 Aug, 10:17 UTC | 35.498°N – 12.621°E 20 ± 2 m asl | 115.9 ± 7.9* | 110.8 ± 4.7 | 5.1 ± 12.6 |
| 10 Aug, 12:14 UTC | 35.500°N – 12.646°E 110 ± 1 m asl | 113.3 ± 6.5* | 109.8 ± 4.7 | 3.5 ± 11.2 |

* Uncertainties are standard deviations

**Table 1.** Comparison of the CO dry volume mixing ratios (vmr) measurements between the WMO-GAW surface station (Picarro instrument) in Lampedusa island (35.52°N-12.63°E, 45 m asl) and the aircraft (SPIRIT instrument).

| Reference | Measurement date | Fire Location | Measurement location | Altitude (km) | Pollutant measured | Transport duration (days) |
|---|---|---|---|---|---|---|
| Formenti et al. (2002) | 14 Aug 1998 | Canada | Aegean Sea | 1-3.5 | Aerosols, CO, $O_3$ | 10 |
| Cristofanelli et al. (2013) | 23-24 Mar 2009 | North America | Italy | - | CO, $O_3$, BC | - |
| Ancellet et al. (2016) | 27-28 June 2013 | Canada & Colorado | Western MB | 2-7 | Aerosols | 10-11* |
| Our study | 10 Aug 2014 | Canada & western USA | Sardinia | 9.7 | CO, $O_3$, BC | 5 |

*10-11-day backward trajectories simulations with FLEXPART

**Table 2.** List of the biomass burning layers coming from the Northern American continent having impacted the MB reported in the literature. The altitude refers to the measurement location.

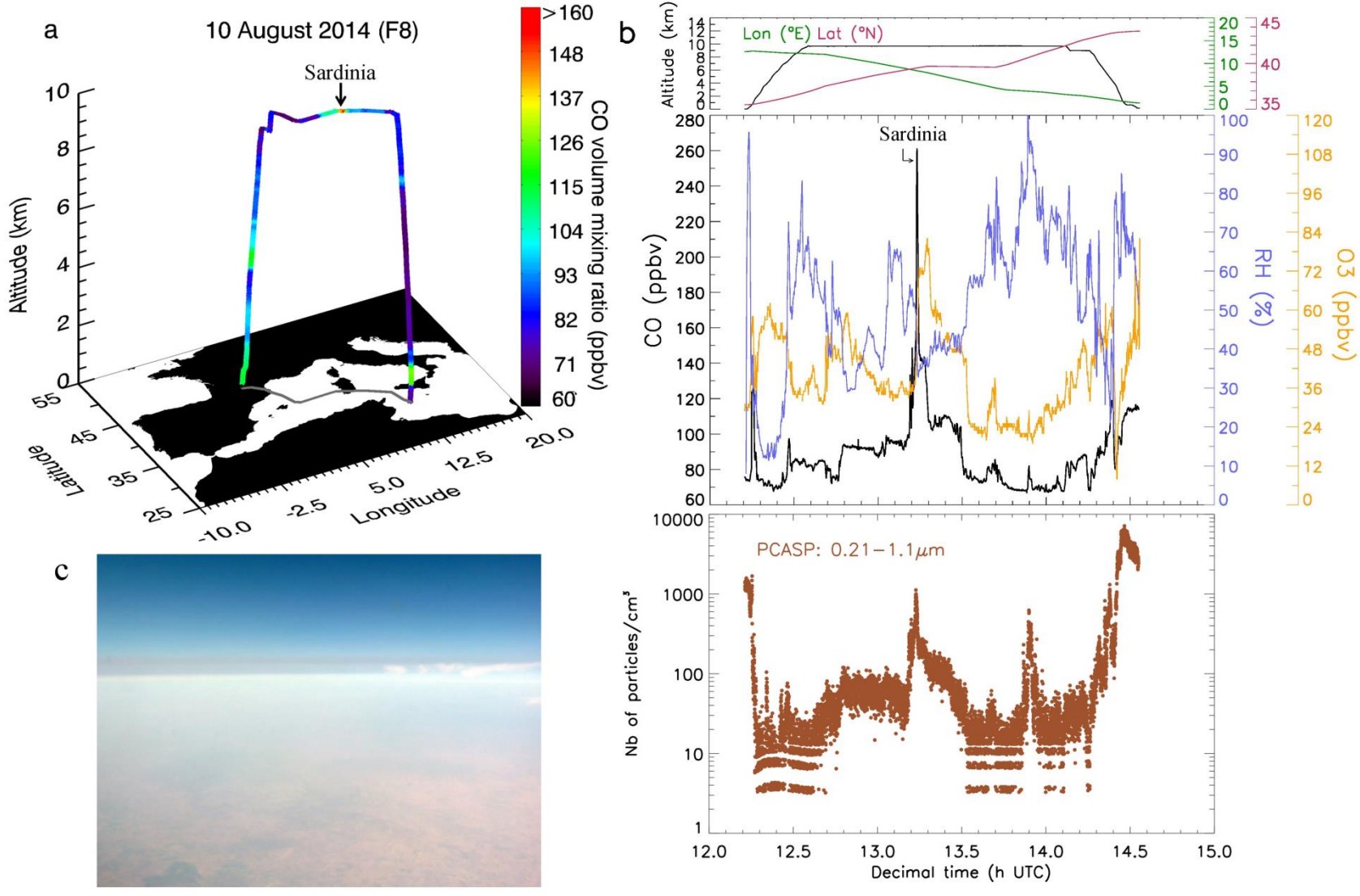

**Figure 1:** Flight F8 (10 August 2014). **(a)** 3D-trajectory color-coded according to CO volume mixing ratios (vmr) between Lampedusa and Toulouse. **(b)** (Top): Flight altitude, longitude and latitude as a function of time; (Middle): Time series of CO vmr (black) and relative humidity (blue); (Bottom): Aerosol total number concentrations (brown). **(c)** Picture of a dark thin layer from the Falcon-20 at an altitude of 9.7 km at 13:12 UTC.

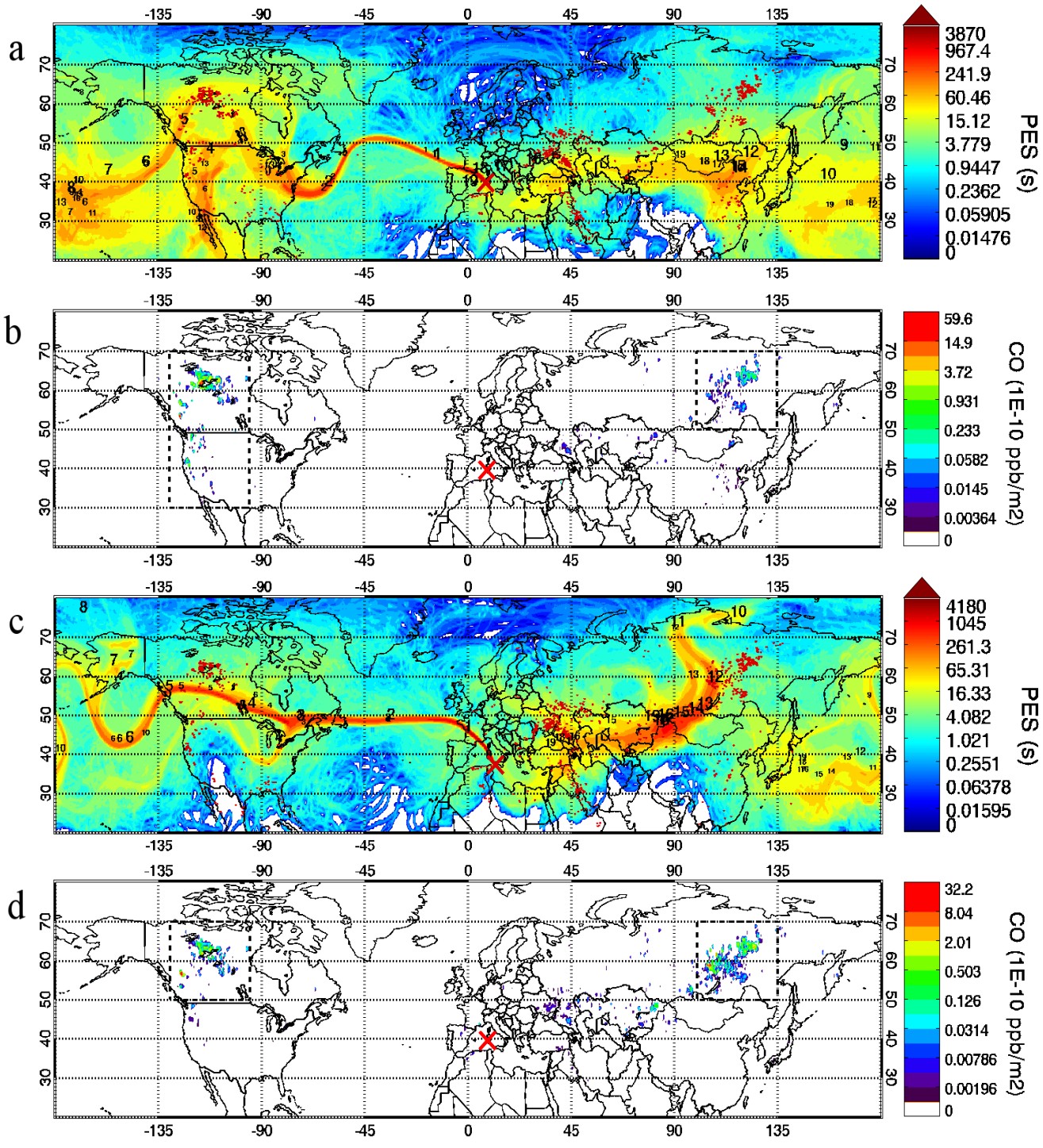

**Figure 2. (a)** and **(c):** Potential emission sensitivity (PES) of the particles in the 0-10 km footprint layer for 20-day backward simulations on 10 August 2014 and 6 August 2014, respectively. Superimposed on (a) and (c) are the MODIS fire detection (dark red dots) and numbers corresponding to the days of back-trajectories with label sizes scaled according to the number of particles belonging to each cluster. The red crosses symbolize the location of the aircraft when measuring the peak of pollution. **(b)** and **(d):** CO biomass burning contribution calculated in the 0-10 km PES layer on 10 August 2014 and 6 August 2014, respectively. The dotted rectangles represent the masks used to see the specific contribution of Siberia and the Northern American continent in Fig. 2b and of Siberia and Canada in Fig. 2d.

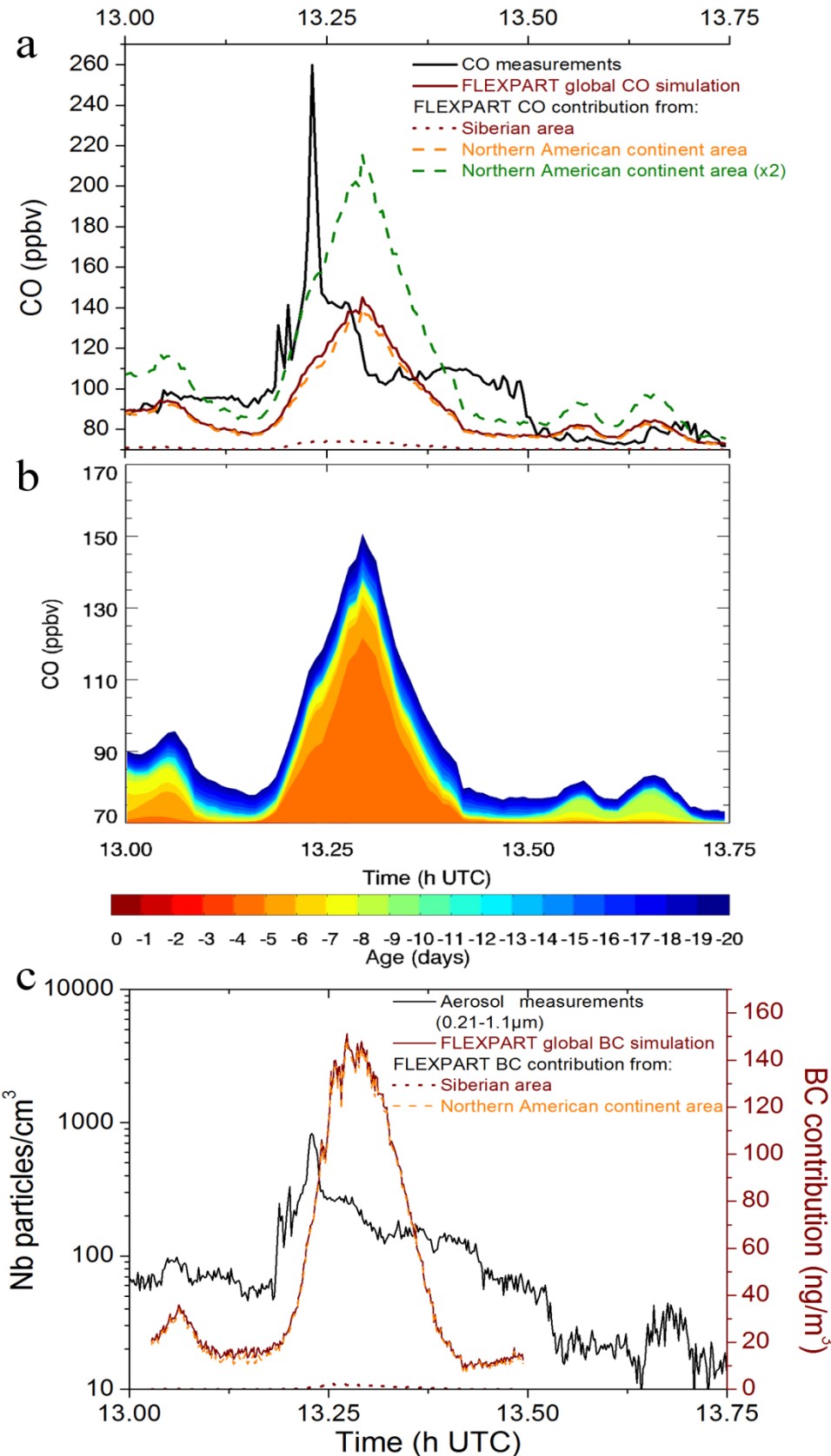

**Figure 3.** Zoom on a part of F8 from 13.0 to 13.75 h UTC above Sardinia at 9.7 km asl with an injection height up to 10 km in FLEXPART. **(a)** CO time evolution along the flight: SPIRIT measurements (degraded to FLEXPART time resolution) and FLEXPART simulations. **(b)** CO contributions calculated by FLEXPART, color-coded according to the age, from 1 to 20 days before F8, of the air mass. **(c)** BC time evolution along the F8 flight: PCASP measurements (degraded to FLEXPART time resolution) and FLEXPART simulations.

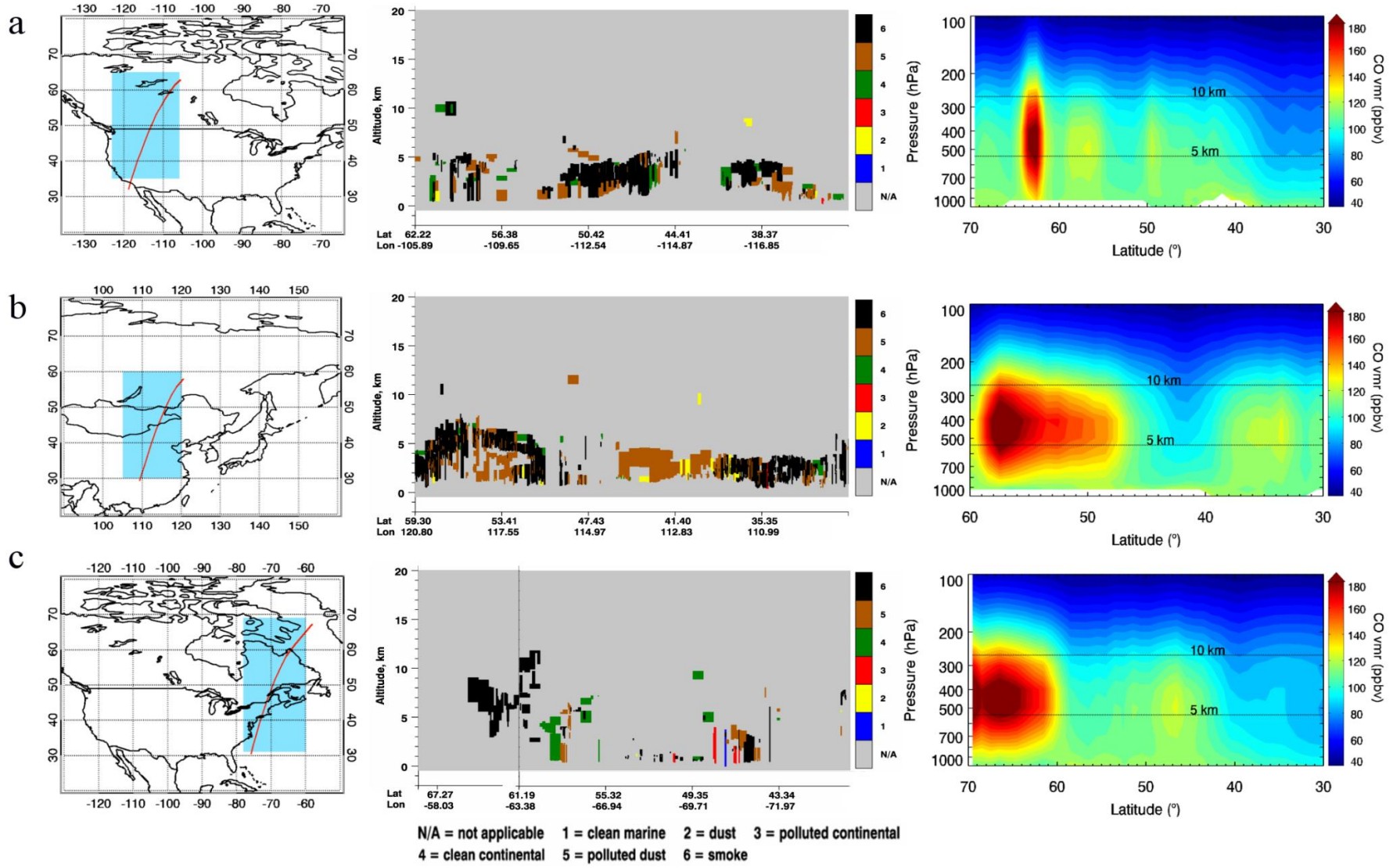

**Figure 4. (a)** 6 August 2014 at 8:51 UTC; **(b)** 24 July 2014 at 18:37 UTC; **(c)** 2 August 2014 at 6:58 UTC. (Left) CALIOP overpass (red line) and area of CO vmr averaging (blue box) from AIRS. (Middle) Vertical distribution of aerosol subtypes (smoke in black) associated to the red line of the CALIOP orbit. (Right) CO vertical distribution from AIRS with data averaged according to the location of the blue box.

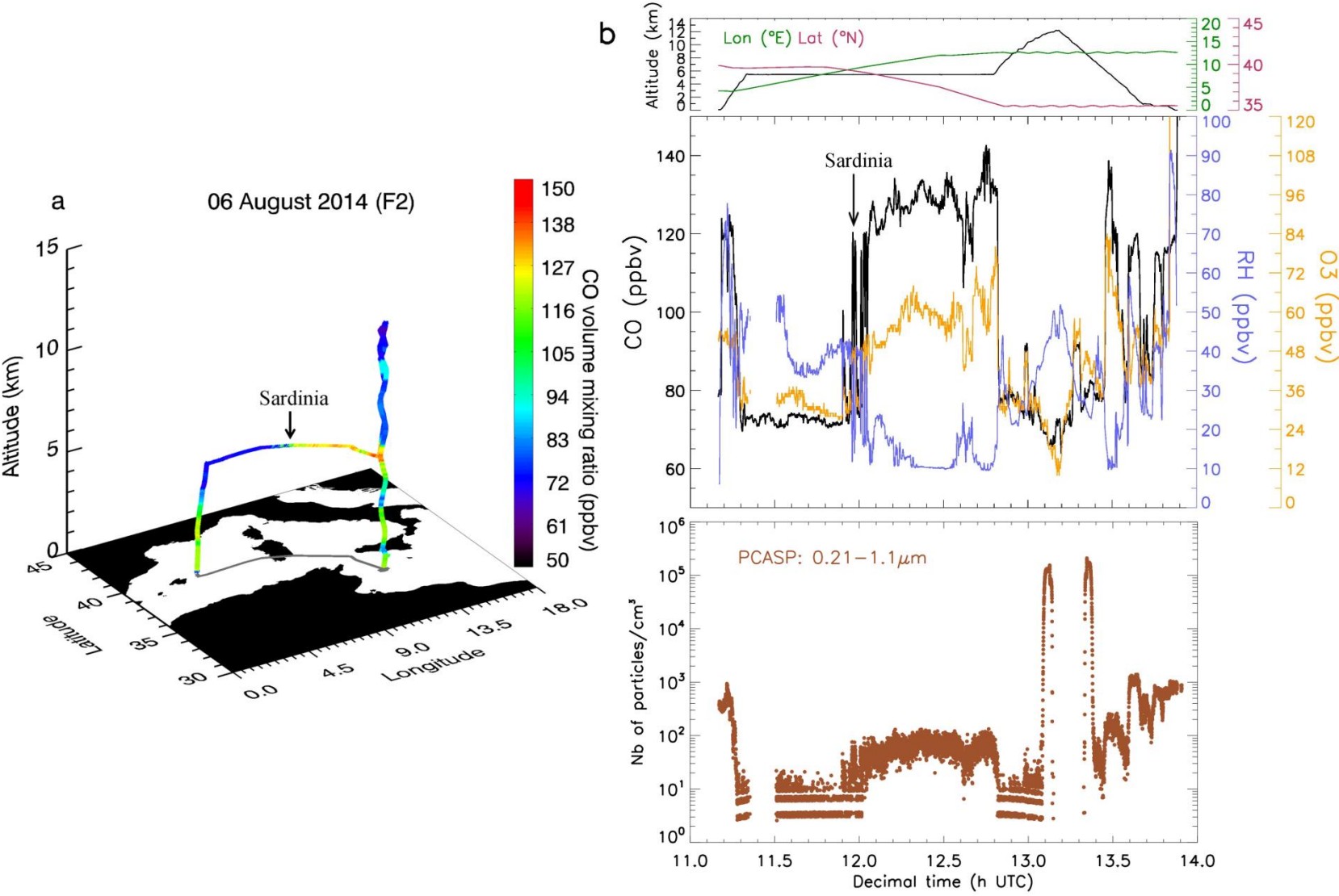

**Figure 5.** Flight F2 (06 August 2014). **(a)** 3D-trajectory color-coded according to CO vmr between Menorca and Lampedusa. **(b)** (Top): Flight altitude, longitude and latitude as a function of time; (Middle): Time series of CO vmr (black) and relative humidity (blue); (Bottom): Aerosol total number concentrations (brown).

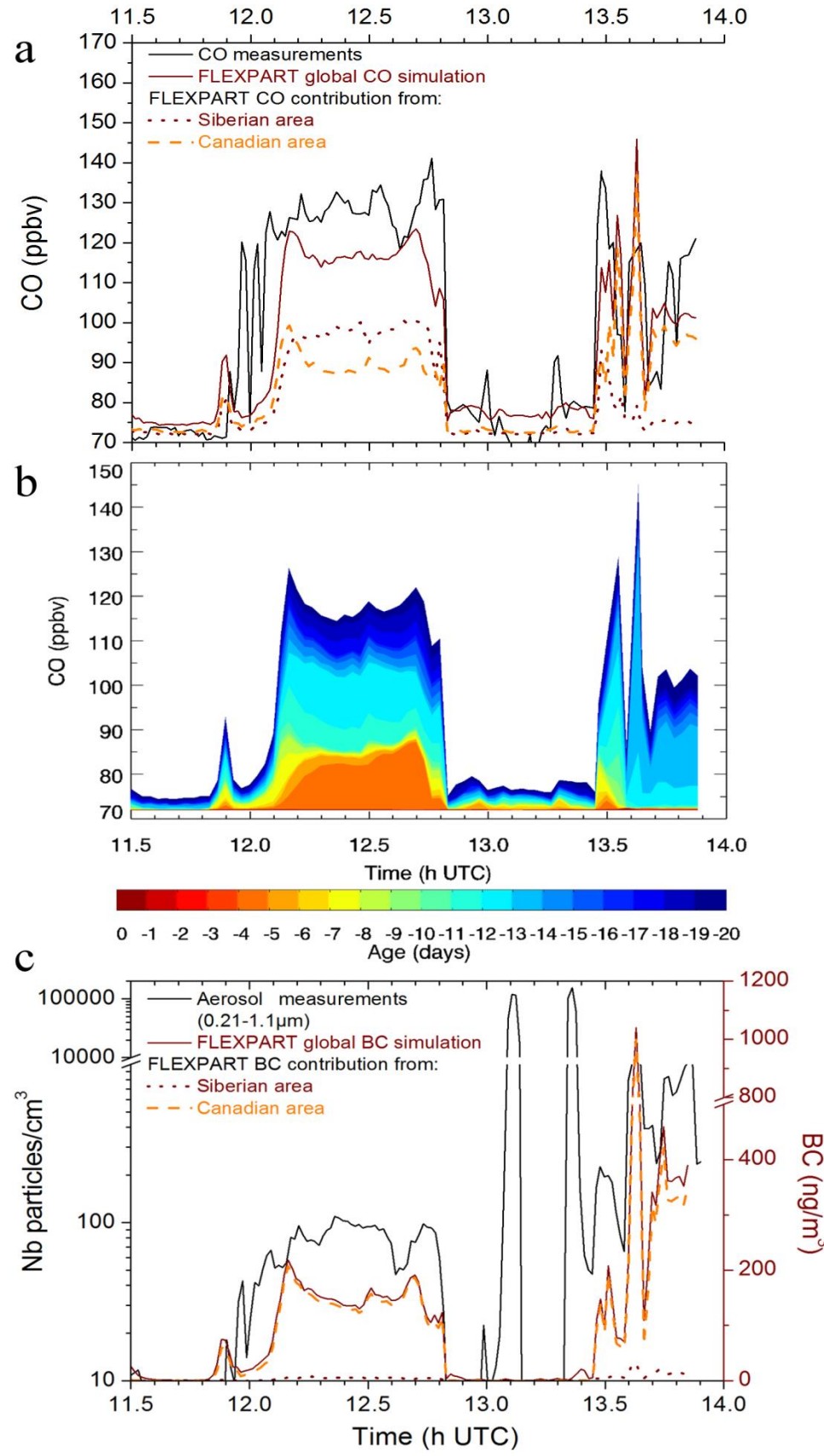

**Figure 6.** Zoom on a part of F2 from 11.5 to 14.0 h UTC above Sardinia at 5.4 km with an injection height up to 10 km in FLEXPART. **(a)** CO time evolution along the flight: SPIRIT vmr (degraded to FLEXPART time resolution) and FLEXPART simulations. **(b)** CO vmr calculated by FLEXPART, color-coded according to the age, from 1 to 20 days before F2, of the air mass. **(c)** BC time evolution along the flight: PCASP measurements (degraded to FLEXPART time resolution) and FLEXPART simulations.

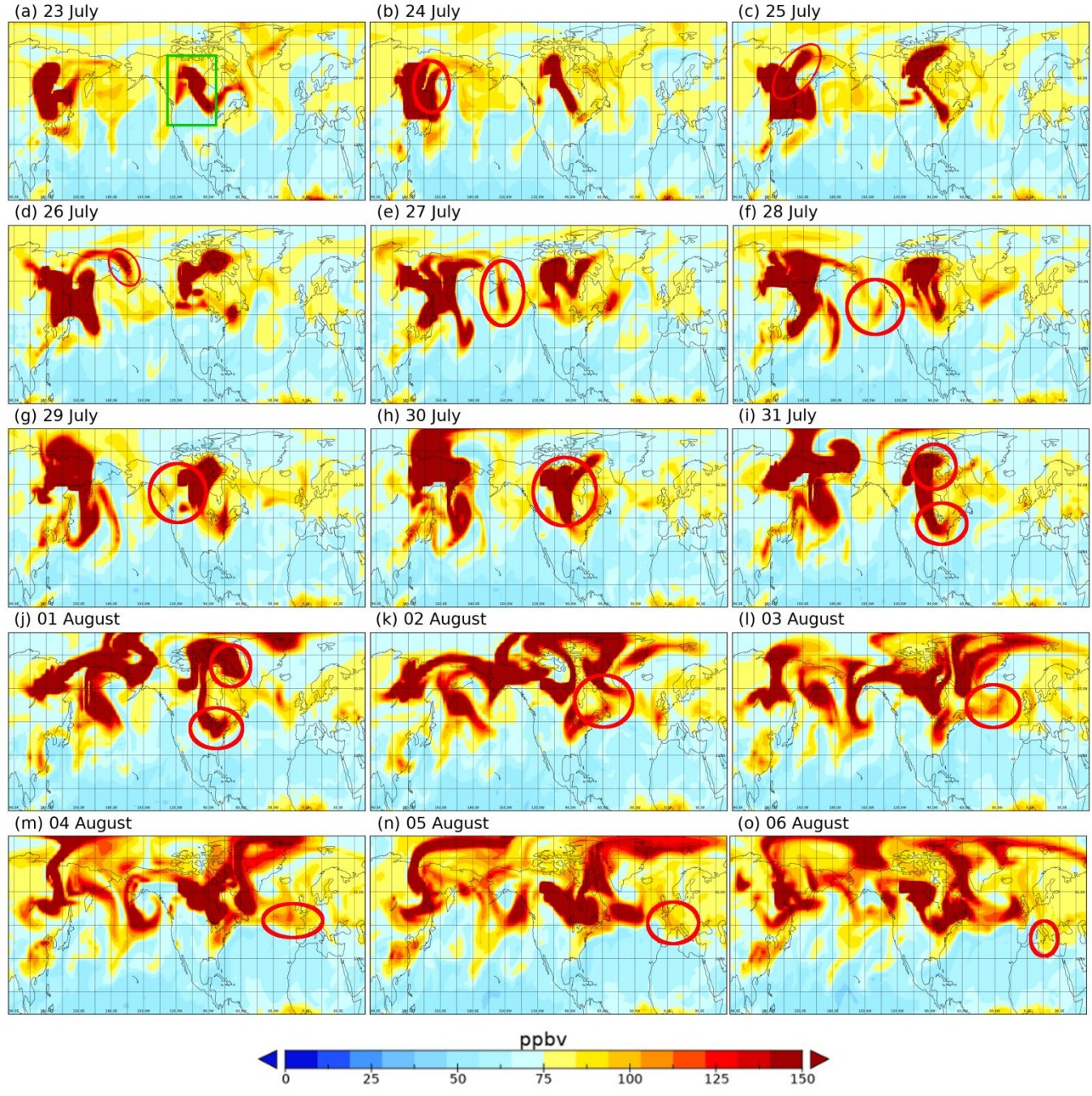

**Figure 7.** CO concentrations at 12 UTC between 23 July and 6 August 2014 at 5.5 km in altitude simulated by the MOCAGE model. The red ellipses are used to follow the biomass burning trace in CO along the trajectory. The green square in (a) corresponds to the box used for Figure 8.

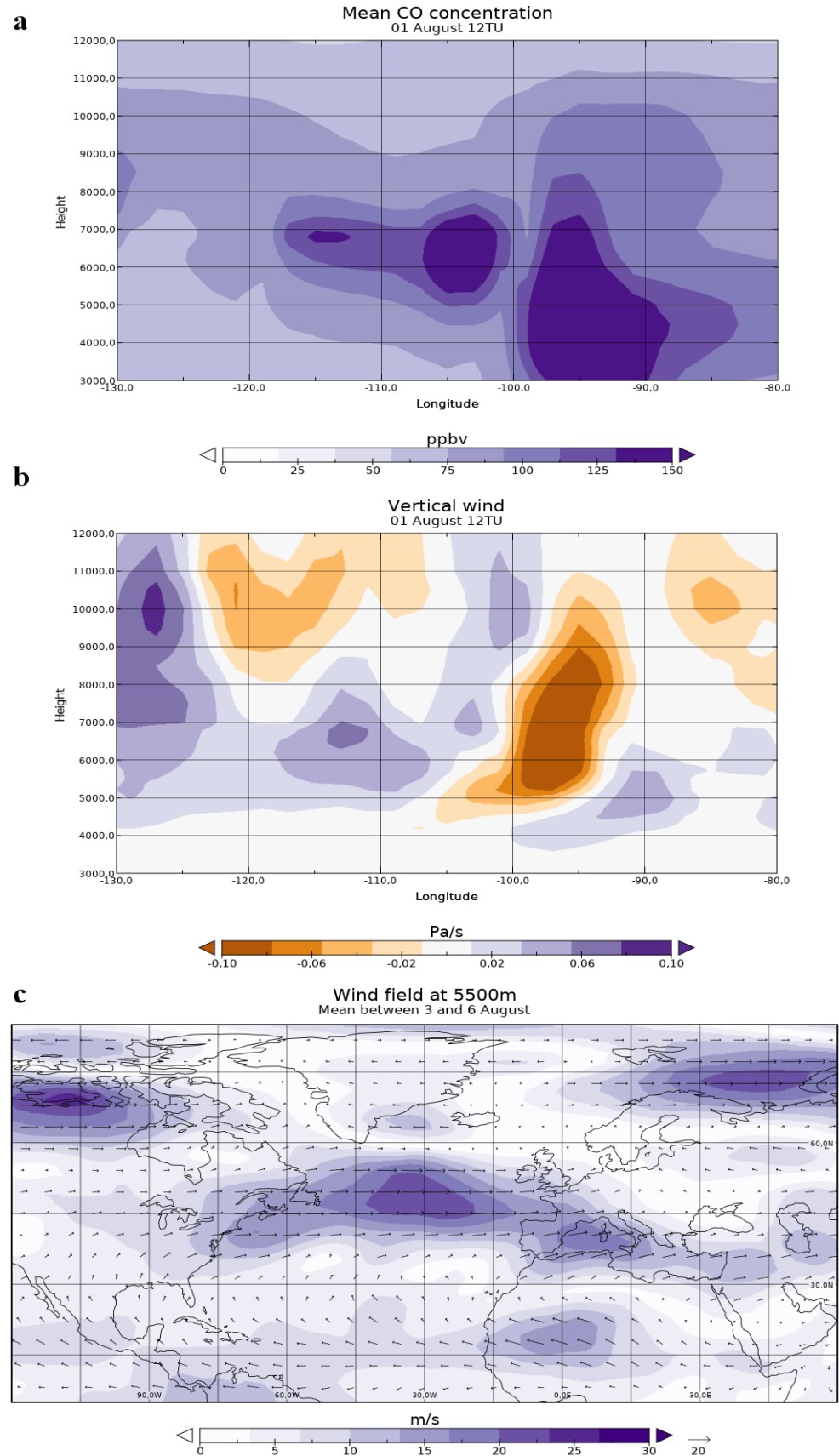

**Figure 8. (a)** Longitude-altitude cross section of the mean CO concentrations from MOCAGE on 1 August, 12:00 UTC, between 40°N and 70°N. **(b)** Vertical wind between 40°N and 70°N (box represented by the green square on Figure 7a) on 1 August 12:00 UTC. **(c)** Mean wind field at 5.5 km asl between 3 and 6 August.

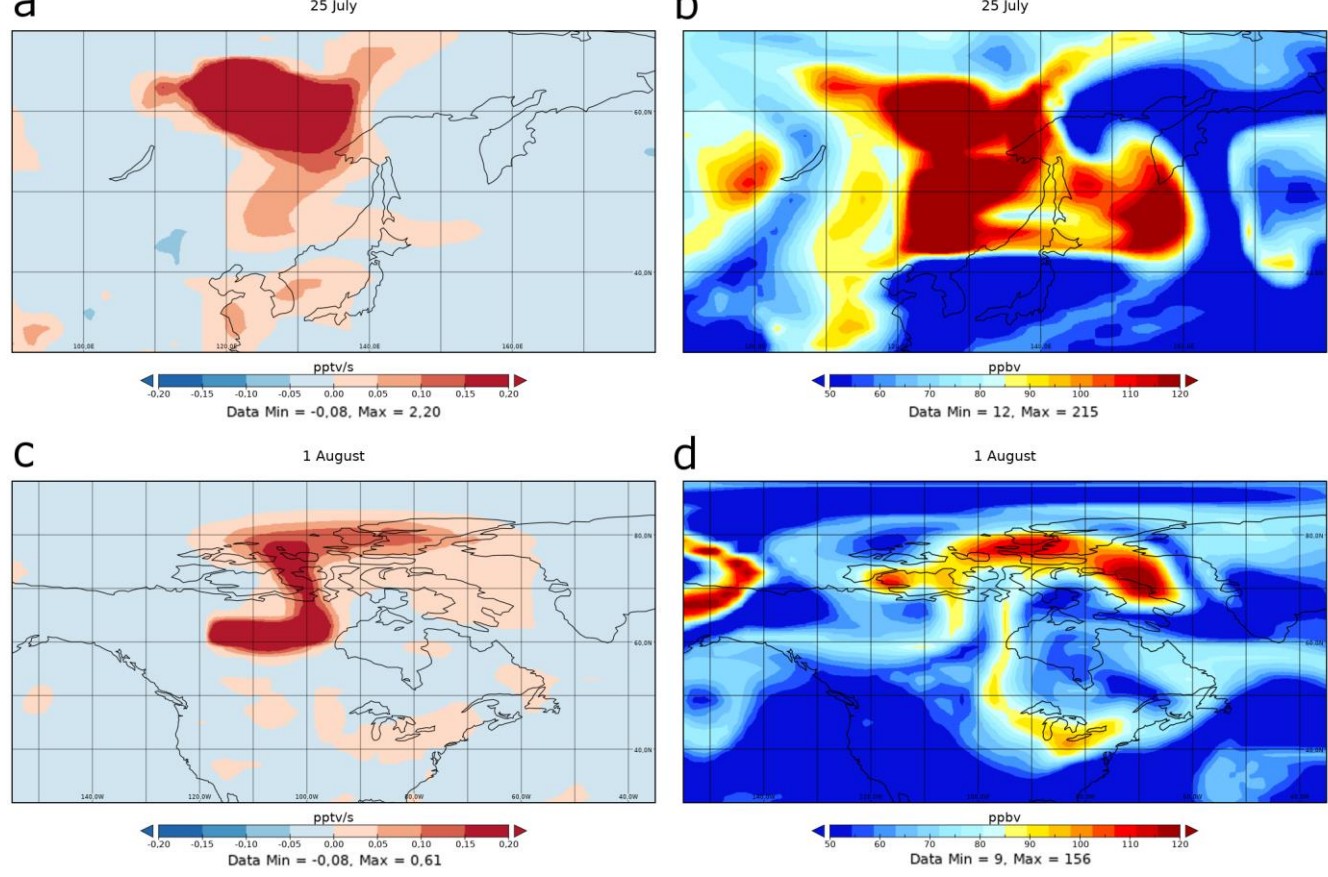

**Figure 9. (a and c)** Production of O$_3$ (in pptv s$^{-1}$) and **(b and d)** concentrations of O$_3$ (in ppbv) on 25 July over Siberia (upper panels) and 1 August 2014 over North America (lower panels) at 5.5 km in altitude simulated by the MOCAGE model.