# Peer review of "Intercontinental transport of biomass burning pollutants over the Mediterranean Basin during the summer 2014 ChArMEx-GLAM airborne campaign"

_Atmospheric Chemistry and Physics, 2017_

## Referee Comment (RC1) · Anonymous Referee #1 · 15 Nov 2017

I found the manuscript submitted by Brocchi et al. very readable and generally clear. It adds evidence of the contribution of hemispheric transport to the atmospheric composition over the Mediterranean basin. I suggest publication of this paper after minor revision, specifically after addressing the following points:

1. The methodology of the study is centred on the use of the Lagrangian particle model FLEXPART backward in time, and the related potential emission sensitivity (PES) tool. I found it difficult to understand the details of this calculation, from the description given in section 2.4. Although I understand more details are given probably in other papers,

at least the minimal information to understand the results of this paper needs come clarification. The method is based on the release of particles from the point of interest (peak of concentrations measured from the aircraft, in this case) and moving back in time. The result is illustrated as a map showing the PES quantity, apparently measured in seconds (s), which intuitively suggest the most "visited" places by the particles. It is unclear, however, how the information from the emission inventory is used: is PES calculated as the time spent in any grid point having a non-zero emissions? The author states that the PES quantity is 3-D (from the surface up to 10 km here) but the map is 2-D (lat-lon): is the quantity shown the vertical integral of this PES? If it is a time quantity, perhaps it is the average? The authors are asked to add more details on this calculations, in order to make fully understandable their results.

2. The description of models and data used does not always report the version number. Where the information is missing, please add the version number of the model and the version number, identification code and url from where data (emission inventories, satellite data, etc.) are taken.

3. The meteorological fields used to run FLEXPART are chosen at 0.5° x 0.5° resolution. In section 2.3 it is however mentioned that the same dataset (ERA-Interim) is used at 0.125° x 0.125°. Please add a note why a degraded resolution is used for the FLEXPART simulation.

4. On line 109, "ro-vibrational" is probably "roto-vibrational".

5. At lines 142-143, the authors claim "no significant difference" between aircraft and ground-based CO concentrations. The term "significant" should be accompanied by a statistical measure such as the p-values, derived by a standard statistical test (e.g. t-test or other non parametric tests). I suggest to include this information, or rephrase avoiding the used of the term "significant". For example, it can be just said that the difference is within the measurement uncertainty.

6. On line 165, the resolution of GEOS-Chem OH field is said to be 3° x 5°, but it is

probably 4° x 5°: please check.

7. On line 250, "The map of CO contribution to biomass burning ...", "to" is probably "from".

8. On line 293 and Figure 3a, the authors illustrate a sensitivity test on fire emission intensity from Canada: is the factor of 2 used here within the expected uncertainty of the related fire emission inventory?

9. Also on the "factor of 2" sensitivity test: in Figure 3a the simulated peak of CO mixing ratio is certainly closer to observations, however also the background values outside the peak are increased, and they are higher than the observations. The factor of 2 multiplicative factor seems to be thus unjustified. The model probably does not capture the intensity of the peak, because of low resolution or numerical diffusion. I this suggest to smooth the statements regarding the possible underestimation by a factor of 2 of the fire emission inventory.

10. Figure 1: the caption reports "Time series of aerosol concentrations ...", I would better call them "aerosol total number concentrations".

11. Figure 2: there seems to be significant fire activity also in southern Russia (north of the Black Sea), which may potential contribute to the air masses captured by the aircraft instruments. I would not expect a significant contribution on the episode of August 10, but perhaps it may play a role on that of August 6, since the contribution from Siberia is found to be larger than that from North America in Figure 6. I suggest to briefly discuss it or revise the calculation for the August 6 episode.

12. Figure 6: there are two peaks around time 13.0 and 13.5 in both CO and BC. Those of BC are larger than the signal discussed in the paper (between times 12-13). These peaks are apparently completely unrelated to forest fires, because are not minimally reproduced by FLEXPART. I suggest to add a note on these peaks in the text, perhaps leaving them for future study or suggesting some speculative hypothesis on their origin

(anthropogenic?).

---

## Referee Comment (RC2) · Anonymous Referee #2 · 30 Nov 2017

This paper reports two interceptions of smoke plumes over the Mediterranean Sea by a French research aircraft, with various chemical sensors, from fires located in North America and Siberia. Two constituents were measured or inferred, black carbon aerosol and carbon monoxide gas. The data was modelled using a well-known trajectory model (FLEXPART) and a less well-known chemistry-transport model (MOCAGE). The data show that emissions from fire plumes on two separate continents can spread widely around the atmosphere, on occasion completing circumnavigation. The discussion of the data is somewhat qualitative and various estimates are made of injection

height from the sources of the plumes and of the amount of carbon monoxide which is released to account for the concentrations detected at the interception point. The reason for the need for this is not discussed any detail or whether this is typical when modelling smoke plumes. The paper is well written and is easy to comprehend. It could be improved in the following ways: 1. In addition to carbon monoxide and black carbon the paper reports that ozone measurements were made on board the aircraft but no use is made of these measurements. This is a major omission. Many papers do comment on ozone production during long-range transport. The authors are aware of this and quote suitable references. 2. It would be easier to understand the vertical structure of the smoke plumes if simple vertical profiles were shown rather than the complex system adopted by the authors with colour coding. The description in the text focuses on horizontal information whereas vertical information would be just as useful since this would indicate the thickness of the layers in a more obvious form. 3. The paper focuses on the use of the trajectory model FLEXPART to identify the origin of the smoke plumes. It does however also refer to the use of the chemistry-transport model but this is only used to confirm the FLEXPART findings. It is not used to comment on any chemistry which may occur as the plume progresses around the atmosphere. Surely some comments regarding ozone production or destruction in the plumes could have been discussed. 4. A minor point: The authors state in the text that on Flight 8 CO reaches 260ppb and the particle count spikes to approximately 1000 particles per ml. The majority of concentrations intercepted on Flight 8 and Flight 2 are rather similar and the higher concentration experienced on Flight 8 are only transitory. The text does not seem to convey this message. 5. On Flight 2, in Figures 5 and 6, two large spikes of particles are shown around 1300 UTC, however there seems to be no increase in CO. There is no comment about this; presumably they are not associated with the fire plumes. Do they contain black carbon for instance? The editors should decide whether the paper makes a sufficient contribution to current knowledge to merit its publication in ACP.

---

## Author Comment (AC1) · 2 Feb 2018

**Manuscript title:** Intercontinental transport of biomass burning pollutants over the Mediteranean Basin during the summer 2014 ChArMEx-GLAM airborne campaign by Brocchi et al.

**RESPONSES TO THE ANONYMOUS REFEREE #1**

We first thank the reviewer for his thoughtful comments that were helpful in improving the manuscript. Changes have been made in response to his specific comments listed below (in black). Our responses appear in red, changes in the revised manuscript in italic.

R. Zbinden has been added to the co-authors due to her participation in the campaign and her collaboration in the data analyses.

1. The methodology of the study is centred on the use of the Lagrangian particle model FLEXPART backward in time, and the related potential emission sensitivity (PES) tool. I found it difficult to understand the details of this calculation, from the description given in section 2.4. Although I understand more details are given probably in other papers, at least the minimal information to understand the results of this paper needs some clarification. The method is based on the release of particles from the point of interest (peak of concentrations measured from the aircraft, in this case) and moving back in time. The result is illustrated as a map showing the PES quantity, apparently measured in seconds (s), which intuitively suggest the most "visited" places by the particles. It is unclear, however, how the information from the emission inventory is used: is PES calculated as the time spent in any grid point having a non-zero emissions? The author states that the PES quantity is 3-D (from the surface up to 10 km here) but the map is 2-D (lat-lon): is the quantity shown the vertical integral of this PES? If it is a time quantity, perhaps it is the average? The authors are asked to add more details on these calculations, in order to make fully understandable their results.

→ It is true that more details are given in the papers cited. However, more explanations are added to the manuscript. One has to distinguish two available products with FLEXPART. The first product is a residence time of the particles in the total atmospheric column while the second one is the PES in a footprint layer (with the altitude of the layer chosen according to the study done: a few hundred meters for anthropogenic emissions and a few km for biomass burning for instance). It is commonly accepted to look at the residence time of the particles in the total column to visualize the most "visited" places (having it in seconds makes it more readable) while the PES (in $s.m^3.kg^{-1}$) coupled with the emission inventories ($kg.m^{-2}.s^{-1}$) gives access to the contribution of the emissions in a specific layer (Stohl et al., 2003). It appears that, in our case, which deals with pyroconvective biomass burning, the footprint layer is more or less equivalent to the total column.

The PES quantity is 3-D but the map presented in the manuscript shows the vertical integrated values (from 0 to 10 km). For each day of simulations, the FLEXPART calculations are daily averaged. Sentences are modified or added in the manuscript giving more details, as follows:

> *This backward mode gives access to two products. The first one is a residence time of the particles in the total atmospheric column while the second one, defined as the potential emission sensitivity (PES), informs on the location where the sampled air mass has been impacted by the surface emissions.*

> *The thickness of the PES layer is chosen consistently with the altitude (vertically integrated values from 0 to 10 km).*

*Outputs are averaged every 24 hours with a horizontal resolution of 0.5°×0.5° globally.*

2. The description of models and data used does not always report the version number. Where the information is missing, please add the version number of the model and the version number, identification code and url from where data (emission inventories, satellite data, etc.) are taken.

→ The requested information has been added. The version R2.15.0 for MOCAGE is specified. The inventories of emissions used in MOCAGE can be found at eccad.sedoo.fr. GFAS data v1.2 can be found at http://apps.ecmwf.int/datasets/data/cams-gfas/. Measurements from AIRS instrument can be found at https://giovanni.gsfc.nasa.gov/giovanni/.

3. The meteorological fields used to run FLEXPART are chosen at 0.5° x 0.5° resolution. In section 2.3 it is however mentioned that the same dataset (ERA-Interim) is used at 0.125° x 0.125°. Please add a note why a degraded resolution is used for the FLEXPART simulation.

→ The time of calculation would be too long with a resolution of 0.125° at global scale (considering the capacity of our computers, of standard quality). However, in section 2.3, studies have been performed at local scale and only wind direction has been visualized. This is the reason why the highest horizontal resolution available with ECMWF has been used. We modified the text as follows:

*Wind fields from ERA-INTERIM are 1) extracted at 6-hourly intervals (0000, 0600, 1200, 1800 UTC) and 3-hour forecasts (0300, 0900, 1500, 2100 UTC) with a horizontal resolution of 0.125°×0.125°, high enough to specify the wind direction at the local scale, and 2) selected at the dates of the presence of the aircraft in the vicinity.*

*Data are extracted at 6-hourly intervals (0000, 0600, 1200, 1800 UTC) and 3-hour forecasts (0300, 0900, 1500, 2100 UTC) with a resolution of 0.5°×0.5° in latitude and longitude, a compromise at the global scale between computing cost and the trajectory accuracy.*

4. On line 109, "ro-vibrational" is probably "roto-vibrational".

→ It is not a typo. The term "Ro-vibrational" is also currently used and admitted in the spectroscopy community.

5. At lines 142-143, the authors claim "no significant difference" between aircraft and ground-based CO concentrations. The term "significant" should be accompanied by a statistical measure such as the p-values, derived by a standard statistical test (e.g. t-test or other non parametric tests). I suggest to include this information, or rephrase avoiding the used of the term "significant". For example, it can be just said that the difference is within the measurement uncertainty.

→ As suggested by the reviewer, we do not use "significant" anymore, and follow the recommendation to use "*the difference is within measurement uncertainty*" in the following rewritten sentence:

*the difference (-3.5 to +5.1 ppb) observed is within the total estimated uncertainties reported for both instruments (4.7 to 7.9 ppb).*

6. On line 165, the resolution of GEOS-Chem OH field is said to be 3° x 5°, but it is probably 4°x 5°: please check.

→ The technical note for FLEXPART (v8.2, http://flexpart.eu/downloads/26) explains that: «A monthly averaged 3°×5° resolution [OH] field averaged to 7 atmospheric levels is used». But it is true that the resolution given in Bey et al. (2001) is 4°×5°. As we do not know whether it is a typo error or a new interpolation of the [OH] field, we have decided to change the reference in the manuscript and give the one of the technical note, downloadable at http://flexpart.eu/downloads/26.

7. On line 250, "The map of CO contribution to biomass burning ...", "to" is probably "from".

→ Yes, we corrected accordingly.

8. On line 293 and Figure 3a, the authors illustrate a sensitivity test on fire emission intensity from Canada: is the factor of 2 used here within the expected uncertainty of the related fire emission inventory?

→ Validation of fire emission inventories is really difficult. We got in touch with Johannes Kaiser, lead author of the GFAS paper (Kaiser et al., 2012), who indicated that the CAMS validation report (Eskes, H. J., et al.: Validation report of the CAMS near-real-time global atmospheric composition service: December 2016 – February 2017, Copernicus Atmosphere Monitoring Service (CAMS) Report, CAMS84_2015SC2_D84.1.1.7_2017DJF_v1.pdf, 2017) gives errors of 30-50% generally for individual fire events. If we apply this range of errors in our case, we are therefore within the expected uncertainty of GFAS inventory. The following sentence has been added:

*As errors depend on individual fire events because the fire space-borne observations depend on the instrument sampling, as e.g. cloud-free observations (J. Kaiser, pers. com.), the underestimation detected in our study cannot be considered as a general statement that has to be applied in every case study using GFAS CO emissions, but only in ours.*

9. Also on the "factor of 2" sensitivity test: in Figure 3a the simulated peak of CO mixing ratio is certainly closer to observations, however also the background values outside the peak are increased, and they are higher than the observations. The factor of 2 multiplicative factor seems to be thus unjustified. The model probably does not capture the intensity of the peak, because of low resolution or numerical diffusion. I this suggest to smooth the statements regarding the possible underestimation by a factor of 2 of the fire emission inventory.

→In line with point 8., we smoothed our statements by mentioning that errors are dependent on each fire and thus, this multiplicative factor is not a factor that has to be applied globally to GFAS emissions. It only corresponds to our results on a particular case study.

10. Figure 1: the caption reports "Time series of aerosol concentrations ...", I would better call them "aerosol total number concentrations".

→ The caption has been rephrased accordingly. It has also been modified in the caption of Figure 5.

11. Figure 2: there seems to be significant fire activity also in southern Russia (north of the Black Sea), which may potential contribute to the air masses captured by the aircraft instruments. I would not expect a significant contribution on the episode of August 10, but perhaps it may play a role on that of August 6, since the contribution from Siberia is found to be larger than that from North America in Figure 6. I suggest to briefly discuss it or revise the calculation for the August 6 episode.

→ New simulations were performed in order to estimate the contribution of the fire activity in Southern Russia (north of the Black Sea) both on the 10 and 6 August (Figure 10 below). The CO

contribution is calculated in an area defined from 25°E to 40°E and 40°N to 55°N. In both cases, it appears that the contribution of CO from biomass burning located in this area is close to zero and, thus has no impact on our results. A sentence has been added in the manuscript but the Black Sea contribution has not been shown in the Figure presented in the revised manuscript:

*The biomass burning contribution in Southern Russia (North of the Black Sea) has also been studied but leads to a contribution close to zero (not shown).*

[Figure]

[Figure]

Figure 10. (**A**) Same as Figure 3a in the manuscript with the contribution of CO from biomass burning calculated in the Southern Russia area, named Black Sea contribution in the figure. (**B**) Same as Figure 6a in the manuscript with the contribution of CO from biomass burning calculated in the Southern Russia area, named Black Sea contribution in the figure.

12. Figure 6: there are two peaks around time 13.0 and 13.5 in both CO and BC. Those of BC are larger than the signal discussed in the paper (between times 12-13). These peaks are apparently completely unrelated to forest fires, because are not minimally reproduced by FLEXPART. I suggest to

add a note on these peaks in the text, perhaps leaving them for future study or suggesting some speculative hypothesis on their origin (anthropogenic?).

→ A coupling between the PES and EDGAR inventory (for anthropogenic emissions) has been performed in order to evaluate the CO contribution from anthropogenic emissions to those two peaks (Figure 11 below). The Figure shows that anthropogenic emissions contribute to about 4 ppbv (compared to an excess of CO of about 15 ppbv) for the first and second peaks. Thus, it cannot only explain the peaks measured but a background of anthropogenic emissions can contribute to it. A hypothesis for the aerosols peaks could be the presence of dust particles although the size presented in the paper (0.21-1.1 μm) is a bit small for this category. We add a note in the text as follows:

*Two peaks of aerosols are also measured around 13.1h and 13.4h UTC (Fig. 5). FLEXPART simulation shows that these peaks are not related to biomass burning as it is not reproduced by the model (Fig. 6c). These two spikes of aerosols could be considered as being dust particles although the associated size (0.21-1.1 μm) is rather small for this type of particles.*

[Figure]

Figure 11. Global CO anthropogenic contribution in a 0-200 m layer on 6 August for the first peak (upper panel) and second peak (lower panel).

**Other modifications in line with the editor's and referee #2 advices:**

- Figure 3c and Figure 6c have been modified. The legend that was first 'BC measurements' becomes 'Aerosol measurements' as we cannot discriminate the type of aerosols with the PCASP instrument.
- The $O_3$ measurements have been added on Figures 1 and 5 in the paper.

---

## Author Comment (AC2) · 2 Feb 2018

**Manuscript title:** Intercontinental transport of biomass burning pollutants over the Mediteranean Basin during the summer 2014 ChArMEx-GLAM airborne campaign by Brocchi et al.

**RESPONSES TO THE ANONYMOUS REFEREE #2**

We thank the reviewer for his thoughtful comments that were helpful in improving the manuscript. Changes have been made in response to his specific comments listed below (in black). Our responses appear in red, changes in the revised manuscript in italic.

R. Zbinden has been added to the co-authors due to her participation in the campaign and her collaboration in the data analyses.

The discussion of the data is somewhat qualitative and various estimates are made of injection height from the sources of the plumes and of the amount of carbon monoxide which is released to account for the concentrations detected at the interception point. The reason for the need for this is not discussed any detail or whether this is typical when modelling smoke plumes.

→ The estimation of the injection height is not so qualitative since it uses CALIOP data as it is commonly done (e.g. Labonne et al., 2007; Kahn et al., 2008; Ancellet et al., 2016). Carbon monoxide has only been used in the Flexpart model with the aim of confirming the presence of concentrations of CO greater than the background value at high altitudes.

1. In addition to carbon monoxide and black carbon the paper reports that ozone measurements were made on board the aircraft but no use is made of these measurements. This is a major omission. Many papers do comment on ozone production during long-range transport. The authors are aware of this and quote suitable references.

→ We agree and added a new paragraph entitled "4. Analysis of $O_3$ production during long-range transport" in the paper:

*The simultaneous increase of CO and $O_3$ measurements shows the production of $O_3$ inside the plume (Fig. 1 and 5). The ratio $\Delta O_3/\Delta CO$ for the increase of the species with respect to their background values averaged over 20 minutes before and after these increases for F2 and F8, respectively, is of about 0.25 for Flight 8 and of about 0.50 for Flight 2. It has been shown that this ratio increases with the age of the plume (Jaffe and Widger, 2012). For our two flights and for boreal regions, these ratios correspond to a plume age ≥ 5 days (Jaffe and Widger, 2012; Parrington et al., 2013; Arnold et al., 2015). More precisely, the ratio gives an approximate plume age of 6-10 days for F8 and of 13-15 days for F2 (Jaffe and Widger, 2012), in agreement with the age of the air mass calculated with FLEXPART.*

*Jaffe, D. A., Wigder, N. L.: Ozone production from wildfires: A critical review, Atmos. Env., 51, 1-10, ISSN 1352-2310, https://doi.org/10.1016/j.atmosenv.2011.11.063, 2012.*

*Parrington, M., Palmer, P. I., Lewis, A. C., Lee, J. D., Rickard, A. R., Di Carlo, P., Taylor, J. W., Hopkins, J. R., Punjabi, S., Oram, D. E., Forster, G., Aruffo, E., Moller, S. J., Bauguitte, S. J.-B., Allan, J. D., Coe, H., and Leigh, R. J.: Ozone photochemistry in boreal biomass burning plumes, Atmos. Chem. Phys., 13, 7321-7341, https://doi.org/10.5194/acp-13-7321-2013, 2013.*

*Arnold, S. R., Emmons, L. K., Monks, S. A., Law, K. S., Ridley, D. A., Turquety, S., Tilmes, S., Thomas, J. L., Bouarar, I., Flemming, J., Huijnen, V., Mao, J., Duncan, B. N., Steenrod, S., Yoshida, Y., Langner, J., and Long, Y.: Biomass burning influence on high-latitude tropospheric ozone and reactive nitrogen in summer 2008: a multi-model analysis based on POLMIP simulations, Atmos. Chem. Phys., 15, 6047-6068, https://doi.org/10.5194/acp-15-6047-2015, 2015.*

2. It would be easier to understand the vertical structure of the smoke plumes if simple vertical profiles were shown rather than the complex system adopted by the authors with colour coding. The

description in the text focuses on horizontal information whereas vertical information would be just as useful since this would indicate the thickness of the layers in a more obvious form.

→ An estimation of the thickness of the biomass burning has been performed for the 6 August. It uses the measurements done during the vertical profile when descending for landing on Lampedusa (Figure 10, not shown in the manuscript). The layer is about 2.9 km thick. A sentence is added in the text, at the end of section 3.2:

> *The measurements performed during this vertical profile help us determining that the thickness of the layer is 2.9 km.*

[Figure]

Fig. 10: Vertical profiles of CO (black) and $O_3$ (orange) vmr, RH (blue) and particles density (brown) on 6 August. The grey rectangle represents the biomass burning layer.

3. The paper focuses on the use of the trajectory model FLEXPART to identify the origin of the smoke plumes. It does however also refer to the use of the chemistry-transport model but this is only used to confirm the FLEXPART findings. It is not used to comment on any chemistry which may occur as the plume progresses around the atmosphere. Surely some comments regarding ozone production or destruction in the plumes could have been discussed.

→ To fill in this gap, we add a section in the paper dedicated to the analysis of the $O_3$ production inside the plume during its long-range transport using the MOCAGE model. See section "4.2 Analysis of $O_3$ production with MOCAGE" in the paper:

> *In this section, MOCAGE simulation is used to analyse the $O_3$ production inside the biomass burning plume during long-range transport. For flight F2, the emissions are set up to an injection height of 10 km without any coefficient applied to the emissions.*

*MOCAGE simulates fairly well the $O_3$ background that is of ~40 ppbv compared to ~32 ppbv for the measurements (not shown). The simulation reproduces the variability of $O_3$ in good agreement with the measurements. For the first period of interest, between 12.0 h and 12.8 h UTC, MOCAGE simulates an increase of ~25 ppbv $O_3$ compared to ~35 ppbv for the measurements. For the second period of interest, at about 13.5 h UTC, MOCAGE simulates an increase of ~30 ppbv compared to ~50 ppbv for the measurements. Note that MOCAGE provides smoother peaks than the observations because of the finer resolution of the observations compared to the model. Considering this, MOCAGE reproduces well the measurements of flight F2 and is thus used to study the production of $O_3$ along the transport.*

*Figure 9 shows both the $O_3$ vmr and $O_3$ production on 25 July and 1 August at 5.5km in altitude. The complete panels of maps from 23 July to 6 August are provided as supplementary material to follow the production (Fig. S1) and the concentrations (Fig. S2) of $O_3$ during the travel of the air mass from Siberia to the MB. It shows high $O_3$ production in the biomass burning plume up to 3 days after the emission (Fig. S1). After that, the ozone production is lowered indicating an aging of the air mass. On 25 July, the production of $O_3$ is visible above Siberia between 40°N and 70°N (Fig. 9a). Figure 9b shows this production of $O_3$ with concentrations of $O_3$ greater than 110 ppbv in the same area. Then, the air mass crosses the Pacific Ocean before arriving above Canada. On 1 August, the simulation shows the production of $O_3$ between 30°N and 90°N (Fig. 9c). The concentrations of $O_3$ in Figure 9d are more important in this area, especially around 45°N with concentrations up to ~100 ppbv and around 70°N with concentrations up to more than 120 ppbv.*

4. A minor point: The authors state in the text that on Flight 8 CO reaches 260ppb and the particle count spikes to approximately 1000 particles per ml. The majority of concentrations intercepted on Flight 8 and Flight 2 are rather similar and the higher concentration experienced on Flight 8 are only transitory. The text does not seem to convey this message.

→ We agree on this statement and added modifications in the text explaining that the increases in CO and BC measured are approximately of the same range and that the huge peak on Flight 8 is a transitory event. A sentence has been added in the text section 3.1:

*During the transect at about 9.7 km asl, an increase of CO vmr up to ~110 ppbv (from a background at ~70 ppbv) has been measured above Sardinia. A very intense and transitory increase of CO up to about 260 ppbv has been measured among this general increase of CO, correlated with a weaker increase in $O_3$ (from ~35 ppbv to ~75 ppbv) and aerosols up to about 1000 particles cm$^{-3}$ in the 0.21-1.1 μm diameter range, and a decrease in relative humidity (RH).*

and in section 3.2:

*The background concentrations are rather similar to the ones measured during F8, however the peak intensity of CO is lower.*

5. On Flight 2, in Figures 5 and 6, two large spikes of particles are shown around 1300 UTC, however there seems to be no increase in CO. There is no comment about this; presumably they are not associated with the fire plumes. Do they contain black carbon for instance?

→ It is not possible, from our measurements, to discriminate the type of aerosols. But, it is clear that from our simulations, we can discard a contribution from biomass burning. See point 12 of the responses to the Reviewer #1 comments.

**Other modifications in line with the editor's and referee #1 advices:**

- Figure 3c and Figure 6c have been modified. The legend that was first 'BC measurements' becomes 'Aerosol measurements' as we cannot discriminate the type of aerosols with the PCASP instrument.
- The $O_3$ measurements have been added on Figures 1 and 5 in the paper.

---

## Author Response (AR2)

**Manuscript title:** Intercontinental transport of biomass burning pollutants over the Mediteranean Basin during the summer 2014 ChArMEx-GLAM airborne campaign by Brocchi et al.

**RESPONSES TO THE EDITOR**

Changes have been highlighted in yellow in the revised manuscript.

Thank you for your careful answers to the referees comments. There are a two small items left before publication.

2. Figure 6: there are two peaks around time 13.0 and 13.5 in both CO and BC. Those of BC are larger than the signal discussed in the paper (between times 12-13). These peaks are apparently completely unrelated to forest fires, because not minimally reproduced by FLEXPART. I suggest to add a note on these peaks in the text, perhaps leaving them for future study or suggesting some speculative hypothesis on their origin (anthropogenic?).

At which altitudes do peaks around time 13.0 and 13.5 appear? Which reasons lead to the attribution of these peaks to dust? This does not explain the concomitant CO peaks.
- The CO peaks (12.99h and 13.28h) both appear at around 10.1 km. As we have already mentioned in the response to the reviewer #1, the coupling between the PES and EDGAR inventory shows that anthropogenic emissions contribute to about 4 ppbv (compared to an excess of CO of about 15 ppbv) for the first and second peaks. Thus, it cannot completely explain the peaks measured. Two hypotheses are given to explain this result. First, although we use the last release version of EDGAR (v4.2), it may be possible that this version is incomplete. Second, those peaks could also be due to a secondary chemical source that would not be taken into account in the inventory.

The aerosol peaks (13.12h and13.35h) that appear around 11.8 and 8.5 km for the first and second peaks, respectively, are not concomitant with the CO peaks. It has been shown in the paper with FLEXPART simulations that those peaks are not associated with the fire plumes. The hypothesis that consists in attributing those peaks to dust comes from two reasons. First, as CO is not concomitant, we think of a natural source of aerosols. Second, if we look at the size range 1.1-3.1 µm measured by the PCASP instrument, we also find two peaks concomitant with the ones measured in the 0.21-1.1 µm size range (see Figure A). As measurements have already been attributed to dust particles during the GLAM campaign (Ricaud et al., 2018), we assume that those peaks could have a similar origin.

As the CO spikes are too small to be clearly identified, we decide to discuss only about the aerosol peaks in the paper and corrected the paragraph previously added in the paper as:

*Two peaks of aerosols, not concomitant with CO, are also measured around 13.1h and 13.4h UTC (Fig. 5). FLEXPART simulation shows that these peaks are not related to biomass burning as it is not reproduced by the model (Fig. 6c). These two spikes of aerosols could be considered as being dust particles as they also appear in the 1.1-3.1 µm size range measured by the PCASP instrument (not shown).*

[Figure]

**Figure A.** Aerosol total number concentrations measured by the PCASP instrument for the 0.21-1.1 µm size range (green dots) and for the 1.1-3.1 µm size range (purple dots).

Fig. 9. The color code is now big enough, however the legend below the color code is too small.

- The figure 9 has been changed to make the legend bigger and becomes:

[revised manuscript text omitted]